# PheWAS-based clustering of Mendelian Randomisation instruments reveals distinct mechanism-specific causal effects between obesity and educational attainment

Liza Darrous [1,2,3] ✉, Gibran Hemani [4,5], George Davey Smith [4,5] & Zoltán Kutalik [1,2,3] ✉

Mendelian Randomisation (MR) estimates causal effects between risk factors and complex outcomes using genetic instruments. Pleiotropy, heritable confounders, and heterogeneous causal effects violate MR assumptions and can lead to biases. To alleviate these, we propose an approach employing a Phenome-Wide association Clustering of the MR instruments (PWC-MR) and apply this method to revisit the surprisingly large apparent causal effect of body mass index (BMI) on educational attainment (EDU): $\hat{\alpha}$ = −0.19 [−0.22, −0.16]. First, we cluster 324 BMI-associated genetic instruments based on their association with 407 traits in the UK Biobank, which yields six distinct groups. Subsequent cluster-specific MR reveals heterogeneous causal effect estimates on EDU. A cluster enriched for socio-economic indicators yields the largest BMI-on-EDU causal effect estimate ($\hat{\alpha}$ = −0.49 [−0.56, −0.42]) whereas a cluster enriched for body-mass specific traits provides a more likely estimate ($\hat{\alpha}$ = −0.09 [−0.13, −0.05]). Follow-up analyses confirms these findings: within-sibling MR ($\hat{\alpha}$ = −0.05 [−0.09, −0.01]); MR for childhood BMI on EDU ($\hat{\alpha}$ = −0.03 [−0.06, −0.002]); step-wise multivariable MR ($\hat{\alpha}$ = −0.05 [−0.07, −0.02]) where socio-economic indicators are jointly modelled. Here we show how the in-depth examination of the BMI-EDU causal relationship demonstrates the utility of our PWC-MR approach in revealing distinct pleiotropic pathways and confounder mechanisms.

Genome-wide association studies[1] (GWASs) have identified many genetic variants associated with multiple complex phenotypes, aiding us in annotating single nucleotide polymorphisms (SNPs) and their functions, as well as identifying putative causal genes. As sample sizes of GWASs increase, more SNP associations are revealed which improve various downstream analyses such as polygenic score prediction, pathway- and tissue-enrichment, and causal inference[2,3].

Mendelian Randomisation[4,5] (MR), an approach generally applied through the use of genetic variants/SNPs as instrumental variables (IVs) to infer the causal relationship between an exposure or a risk factor $X$ and an outcome $Y$, has become increasingly used thanks to

[1]University Center for Primary Care and Public Health, Lausanne, Switzerland. [2]Swiss Institute of Bioinformatics, Lausanne, Switzerland. [3]Department of Computational Biology, University of Lausanne, Lausanne, Switzerland. [4]Medical Research Council Integrative Epidemiology Unit, Population Health Sciences, University of Bristol, Bristol, United Kingdom. [5]Population Health Sciences, Bristol Medical School, University of Bristol, Bristol, United Kingdom. ✉e-mail: liza.darrous@unil.ch; zoltan.kutalik@unil.ch

well-powered GWASs from which hundreds of genetic associations with heritable exposures can be used as IVs.

MR has three major assumptions concerning the genetic variant $G$ used as an instrument: (1) Relevance – $G$ is strongly associated with the exposure. (2) Exchangeability – there is no confounder of the $G$-outcome relationship. (3) Exclusion restriction – $G$ affects the outcome only through the exposure. Each instrument provides a causal effect estimate, which can then be combined with others using an inverse variance-weighting[6] (IVW) method to obtain an estimate of the total causal effect of the exposure on the outcome. This estimate is more reliable than observational associations due to it being more protected against unmeasured confounding and reverse causality, provided that the core conditions are met.

Thanks to well-powered GWASs, we have also discovered that most genetic instruments are highly pleiotropic[7], i.e. associated with more than a single trait. This has also been shown in phenome-wide association studies (PheWASs), where associations between a SNP and a large number of phenotypes are tested. The situation where a genetic variant influences multiple traits, but there is a primarily associated trait which mediates all other trait associations, is referred to as vertical pleiotropy. On the other hand, genetic variants that affect some traits through pathways other than the primary trait (the exposure) – a phenomena known as horizontal pleiotropy – are in direct violation of the exclusion restriction assumption and could lead to biased causal effect estimates. However, if the InSIDE assumption[8] (Instrument Strength is Independent of the Direct Effect on the outcome) holds and the direct SNP effects are on average null, then IVW will yield consistent causal effect estimates. There have been MR extensions to IVW such as MR-Egger to produce less biased causal effect estimates if the InSIDE assumption holds and direct effects are not null on average. Note that violation of the InSIDE assumption leads to correlated pleiotropy, which can severely bias causal effect estimates. Such a phenomenon may emerge as a result of a heritable confounder of the exposure-outcome relationship and has been modelled in the past[9,10].

Well-powered GWAS may also provide confounded genetic associations through dynastic effects[3,11], assortative mating[12,13], and population stratification[14]. These phenomena can introduce correlation between an instrument and confounding factors, such as parental/partner traits or genetic ancestry, leading to a violation of the exchangeability assumption and biased causal effect estimates. This type of confounding can be resolved when using family-based study designs[15,16] such as sibling-pair studies. Since genetic differences between sibling pairs are due to independent and random meiotic events, these effects are unaffected by population stratification and other potential confounders influencing the phenotype. This and other emerging family-based designs have been used to obtain unbiased heritability estimates, validate GWAS results, and test for unbiased causal effect estimates using MR[17,18].

Another factor that can lead to complications in MR studies is the presence of heterogeneous causal effects emerging due to distinct biological mechanisms: various subtypes of the exposure (e.g. subcutaneous *vs.* visceral adiposity) or different biological pathways through which the exposure impacts the outcome (e.g. interaction between the exposure and other factors). To date, horizontal pleiotropy, confounding of genetic associations, and heterogeneous causal effect have been largely treated as distinct mechanisms in MR modelling. However, what they have in common is that they can lead to variable causal effects estimated depending on the group of IVs used in the MR analysis.

To address this, we introduce in this paper our approach of PheWAS-driven clustering of instrumental variables (PWC-MR) and test the resulting clusters for distinct pathways or mechanisms that could underlie the overall causal effect of the exposure. Throughout the paper, we demonstrate the approach through the example of estimating the causal effect of body mass index (BMI) on educational attainment (EDU). This relationship has been analysed extensively in the past and family studies have shown that an apparent strong effect of higher BMI on lower educational attainment is shrunk to near zero when using family studies[17]. One explanation is that offspring BMI is influenced by parental alleles associated with parental (rearing) behaviour, which in turn modify the environment of the offspring. Such parental traits act as a confounder of the offspring genotype-EDU relationship, hence violating the exchangeability assumption of MR. Moreover, they confound the BMI-EDU association in the tested sample, violating the exclusion-restriction assumption and inducing correlated pleiotropy (see Fig. 1). Thus, it is plausible that some of the detected IV clusters arise through parental genetic confounding which may manifest statistically as horizontal pleiotropy. To test this, we ran a systematic confounder search and probed the causal effect of the exposure conditional on candidate confounder traits.

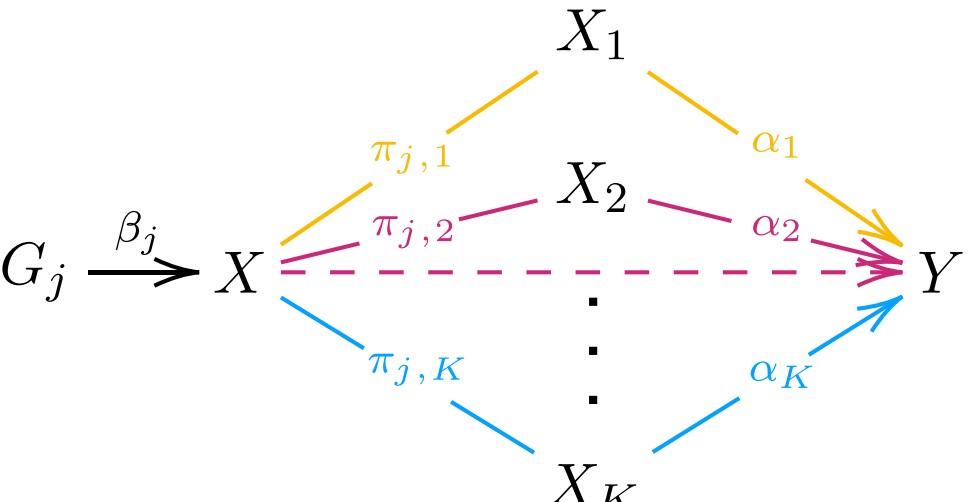

**Fig. 1 | Directed Acyclic Graph (DAG) illustrating the complex relationship between exposure and outcome.** $G_j$ represents genetic instrument $j$ with an effect $\beta_j$ on exposure $X$. Exposure $X$ is associated with outcome $Y$ through $K$ possible pathways of mediation or confounding denoted through the various $X_1...X_K$ elements. The associations between the main exposure and the various elements denoted by the $\pi$ arrows purposely do not show directionality to allow for both mediators and confounders. The causal effects on outcome $Y$ are denoted by $\alpha_1, \alpha_2, \ldots, \alpha_K$.

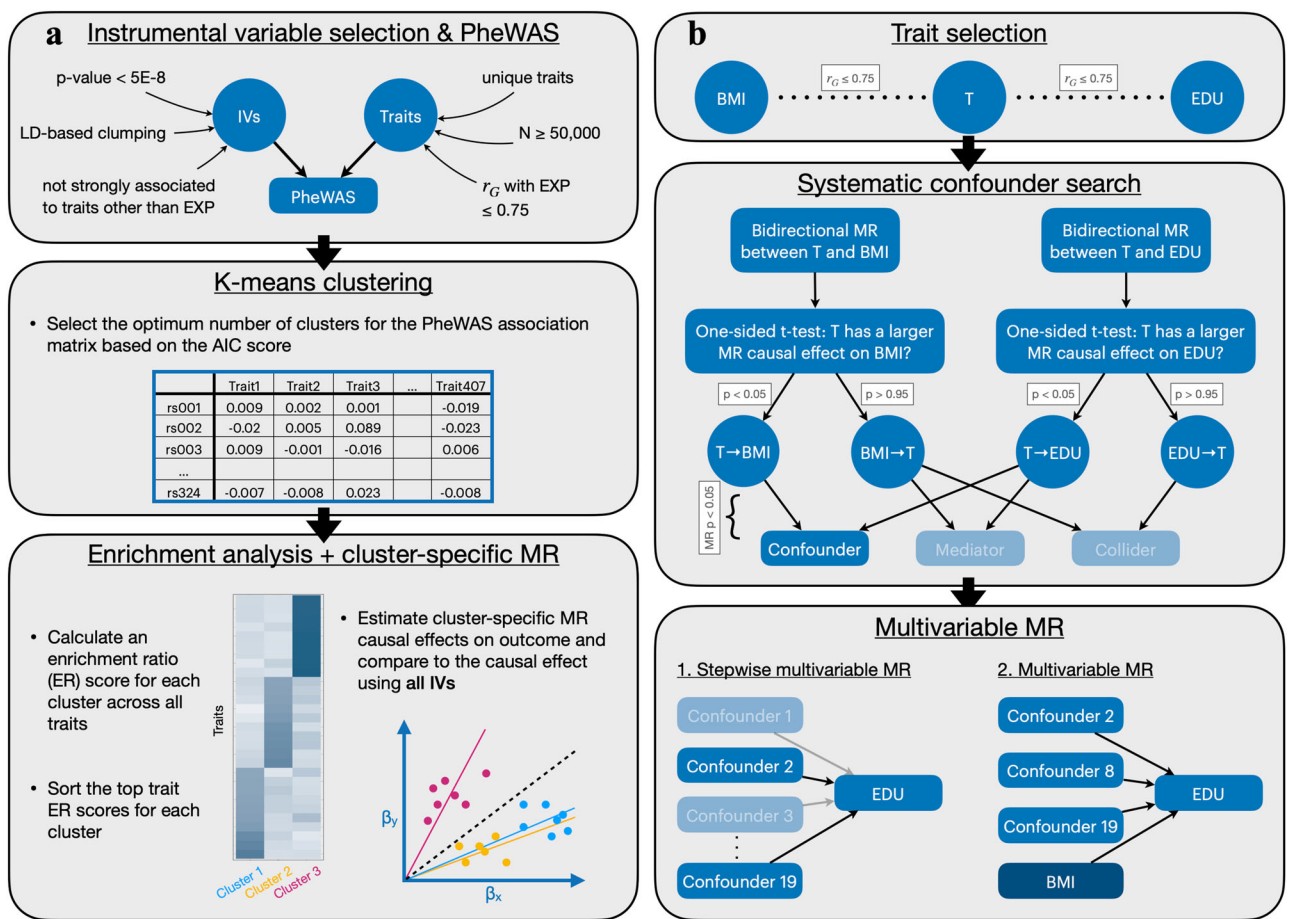

**Fig. 2 | Flow diagram representing how the PWC-MR approach aims to disentangle causal effect between trait pairs from confounding or pleiotropy, as well as systematically search for confounders of the trait pair.** Panel **a** represents the main steps of the PWC-MR method: (i) Instrument selection and PheWAS; (ii) Informative IV clustering using K-means; and (iii) Enrichment analysis and cluster-specific MR. Panel **b** represents a complimentary approach to PWC-MR where a systematic candidate confounder trait search is performed. These candidate confounder traits are defined as having an effect on both the exposure and the outcome. A stepwise multivariable MR (MVMR) of the candidate confounder traits is performed to select those with a strong effect on the outcome. These are then added with the primary exposure (BMI) to a standard MVMR and the multivariable causal effect on the outcome (EDU) is estimated. Acronyms: N: sample size, $r_G$: genetic correlation, T: trait, p: t-test p-value; MR p: MR p-value.

## Results

### Overview of the method

Horizontal/correlated pleiotropy, confounded genetic associations, and mechanism-specific causal effects all lead to heterogeneous MR causal effect estimates. In PWC-MR, we attempt to investigate all these possible biases simultaneously by informatively clustering the various IVs and testing the resulting groups for distinct pathways or mechanisms underlying the overall causal effect as illustrated in Fig. 1.

We applied the PWC-MR approach to investigate potential horizontal pleiotropic effects (emerging due to heritable confounders, dynastic effects, genetic subtypes of obesity and other pleiotropic mechanisms, see Fig. 1) of BMI on educational attainment. The analysis focused on grouping the IVs of the exposure by running a PheWAS-based clustering to reveal distinct mechanisms or pathways underlying their overall effect on the outcome (Fig. 2a). This was done by obtaining the standardised PheWAS association of the BMI IVs across a filtered set of 407 traits, and running a k-means clustering on the resulting matrix. This resulted in six clusters of IVs for BMI, which were then annotated by traits based on the association of the clustered SNPs with each trait. Specifically, for each cluster-trait pair we computed the average explained variance of the trait by the SNPs of the given cluster. This yielded an enrichment ratio (ratio of the average explained variances) for each cluster-trait pair, and we chose the top ten traits with the highest enrichment ratio for each cluster as representatives. Furthermore, the causal effect of each cluster's IVs on education was calculated and compared against each other and that of the causal effect obtained using all BMI IVs.

To complement our findings from the clustering-based analysis, we explored (i) the BMI-EDU causal relationship using sib-regression SNP effect sizes[18], (ii) the childhood BMI-EDU causal relationship, (iii) replacing the outcome trait with systolic blood pressure (SBP), and finally (iv) the potential role of each of the filtered set of traits as a confounder of the BMI-EDU relationship.

We implemented the latter one by systematically running bidirectional MR between each of the traits and either BMI or EDU as outcome, then classifying the traits depending on their bidirectional associations with both BMI and EDU. The resulting set of candidate confounder traits was further analysed for its potential to bias the causal effect of BMI on EDU. To assess this, we ran stepwise MVMR and finally calculated the causal effect of BMI on EDU conditional on the surviving set of candidate confounder traits of the BMI-EDU relationship (illustrated in Fig. 2b).

To further understand the emerging clusters, we sought to uncover tissue-specific mechanisms. To do this, we performed a colocalisation analysis of the BMI and gene expression association signals at each locus around (±400kb) the 324 BMI IVs. For the gene expression association, we used eQTL data from both adipose and

brain tissue. This yielded a proportion of brain-vs-adipose colocalised IVs for each cluster.

### PheWAS-based clustering, annotation and cluster-specific causal effects

After identifying 324 genome-wide significant SNPs as IVs for BMI, and selecting 407 filtered traits to run PheWAS on, we obtained a standardised effect matrix of the 324 IVs on the 407 traits. Normalising the matrix by IVs and running K-means clustering on it revealed that six clusters yielded the lowest AIC score (Supplementary Fig. 1) when compared to varying the number of clusters from two to 50. The number of SNPs in each of the six clusters were: 32, 98, 35, 41, 69, 49 respectively (Supplementary Data 2).

Next, we computed an enrichment ratio (ER) to identify with which traits the SNPs in each cluster were strongly associated. The overall ER value between clusters was roughly centred around 1, however, clusters #2, #3, #4, and #6 had some large ER values (see Supplementary Fig. 2). Visualising the top 10 enriched traits in each cluster and their ER values in Fig. 3 and Supplementary Data 3, we see that cluster #2 is strongly enriched for lean mass traits such as 'Trunk fat-free mass' and 'Whole body fat-free mass'. Similarly, cluster #3 seemed to mostly be enriched for blood- and body stature-related traits such as 'Platelet count' and 'Standing height', while cluster #4 was enriched for traits related to socio-economic position (SEP) such as 'Job involves heavy manual or physical work', 'Time spent outdoors in summer', and 'Fluid intelligence score'. Lastly, cluster #6 was enriched for food supplements/nutrients such as 'Folate' and 'Potassium'.

To test whether the clusters had different causal effects on a selected outcome than the overall causal effect (using all IVs), we computed the IVW causal effect estimate of each cluster on education using cluster-specific IVs. As seen in Fig. 4a and Supplementary Data 4, the causal effect estimates between the different clusters are significantly heterogeneous (Q-test value = 130.61, $p$-value < $10^{-300}$). Clusters #2 and #5 had the smallest causal effect estimates of −0.09 ($p$-value = $1.23 \times 10^{-5}$) and −0.12 ($p$-value = $5.22 \times 10^{-6}$) respectively, where cluster #2 was enriched for lean-mass traits. These estimates are consistent with those obtained from within-family studies, which are relatively immune to confounding (see Sensitivity analyses section below). By contrast, clusters #1 and #4 had the largest negative causal effect estimates of −0.44 ($p$-value = $7.78 \times 10^{-20}$) and −0.49 ($p$-value = $1.63 \times 10^{-44}$) respectively, where cluster #4 was strongly enriched for SEP-related traits. All the clusters were less heterogeneous than the group of all the IVs combined (see 'Avg_het' in Supplementary Data 4).

### Sensitivity analyses

To test the robustness of the PWC-MR results, we performed three additional analyses. First, we used the same exposure and outcome, but the MR analysis was based on sib-regression-based SNP effect sizes instead of SNP effects from GWAS of unrelated samples. Second, we replaced the exposure with childhood BMI and estimated its causal effect on EDU. Lastly, we replaced the outcome, EDU, with SBP.

In Howe et al.[18], within-sibship (within-family) meta-analysed GWAS estimates were generated from 178,086 siblings across 19 cohorts. Using these effect estimates, MR was performed with BMI as exposure on multiple traits, including educational attainment. They used 418 independent and genome-wide significant genetic variants for BMI, and estimated its effect on EDU using IVW to be −0.05 (95% CI: −0.09, −0.01).

They also used jackknife to estimate the standard error of the difference between the sib-regression MR estimate and that of the GWAS of unrelated samples MR estimate: −0.19 (95% CI: −0.22, −0.16). Using the difference Z-score to generate a $p$-value for heterogeneity between the two estimates revealed a significant difference with a $p$-value < 0.001.

We used the UK Biobank trait 'Comparative body size at age 10' as a proxy for childhood BMI – a measure that has been validated against measured BMI in childhood[19,20] – for the exposure trait. Childhood BMI is presumed to be less influenced by SEP compared to adult BMI and hence we expect the causal effect estimate on EDU to have less confounding bias. For this trait, we had 171 genome-wide significant SNPs that we used as IVs for the analysis. Of these, 16 SNPs were more strongly associated to traits other than childhood BMI and were thus excluded from further analysis. The standardised effect matrix of the remaining 155 SNPs across 461 traits was normalised with respect to the SNPs, and then clustered into four clusters (yielding optimal AIC), each containing 37, 42, 32, 44 IVs, respectively (Supplementary Fig. 3, Supplementary Data 5).

Analysing the trait enrichment for each cluster revealed only two clusters with high ER values: clusters #2 and #4 (Supplementary Fig. 4, Supplementary Data 6). Cluster #2 had only two traits with ERs greater than 2, which were 'Number of fluid intelligence questions attempted within time limit' and 'Fluid intelligence score', whereas cluster #4 was highly enriched for body-measurement/fat-mass traits such as 'Waist circumference' and 'Whole body fat mass' (see Supplementary Fig. 5). However, calculating the IVW causal effect estimate for each cluster and comparing it to the estimate calculated using all IVs revealed homogeneous causal effect estimates with a Q-statistic of 3.84 ($p$-value of 0.43) as seen in Fig. 4b and Supplementary Data 7. Cluster #2 had a causal effect estimate of −0.09 (95% CI: −0.1638, −0.0148), and cluster #4 had a causal effect estimate of −0.04 (95% CI: −0.0823, −0.0024). Noteworthy is the finding that the IVs of cluster #2 were more heterogeneous than all the IVs combined. Thus, we obtained a massively attenuated causal effect of BMI on EDU, when childhood BMI is used as an exposure. Reassuringly, no strongly SEP-enriched cluster emerged and the cluster-specific causal effects were homogeneous.

To find further evidence that our approach does not always reveal distinct causal effects when the causal effect is non-null, we replaced EDU with SBP as outcome. Namely, we tested a well-established non-null causal relationship that is hypothesised to not be biased by pleiotropy or confounding: BMI's effect on SBP. Using the same six clusters previously obtained for BMI, we calculated the estimated causal effect of each of the clusters compared to using all the IVs combined on SBP. This revealed a homogeneous set of causal effect estimates (Q-test value of 4.49, $p$-value = 0.61), with the IVW estimate using all IVs being 0.15 ($p$-value = $1.09 \times 10^{-28}$) as seen in Fig. 4c and Supplementary Data 8.

### Systematic confounder search and MVMR analysis

Given our suspicion that the large BMI-EDU causal effect is driven by heritable confounders, we performed a systematic search to reveal traits that may be potential confounders. As described in the Methods section, the strength of the bidirectional effect of the traits on either the exposure or the outcome determined their categorisation. This led to the identification of 19 traits that were found to be candidate confounder traits (Supplementary Data 9). Matching the 19 confounder traits from this analysis to their respective ERs across the six clusters from the previous analysis revealed higher ERs in cluster #1 and cluster #4 (associated with SEP-related traits), both of which also had the largest negative causal effects on EDU (Supplementary Fig. 6).

It is worth noting that the traits labelled as candidate confounders were predominantly environmental exposures, such as 'Exposure to tobacco smoke outside home' and 'Transport type for commuting to job workplace: Cycle'. Furthermore, these candidate confounder traits are attributed as candidate or potential confounders since they are most likely only genetic correlates of the true confounding traits of the BMI-EDU relationship and do not act as true confounders themselves.

To investigate the possible biasing effect that potential confounder traits can have on the causal relationship of BMI on EDU, we

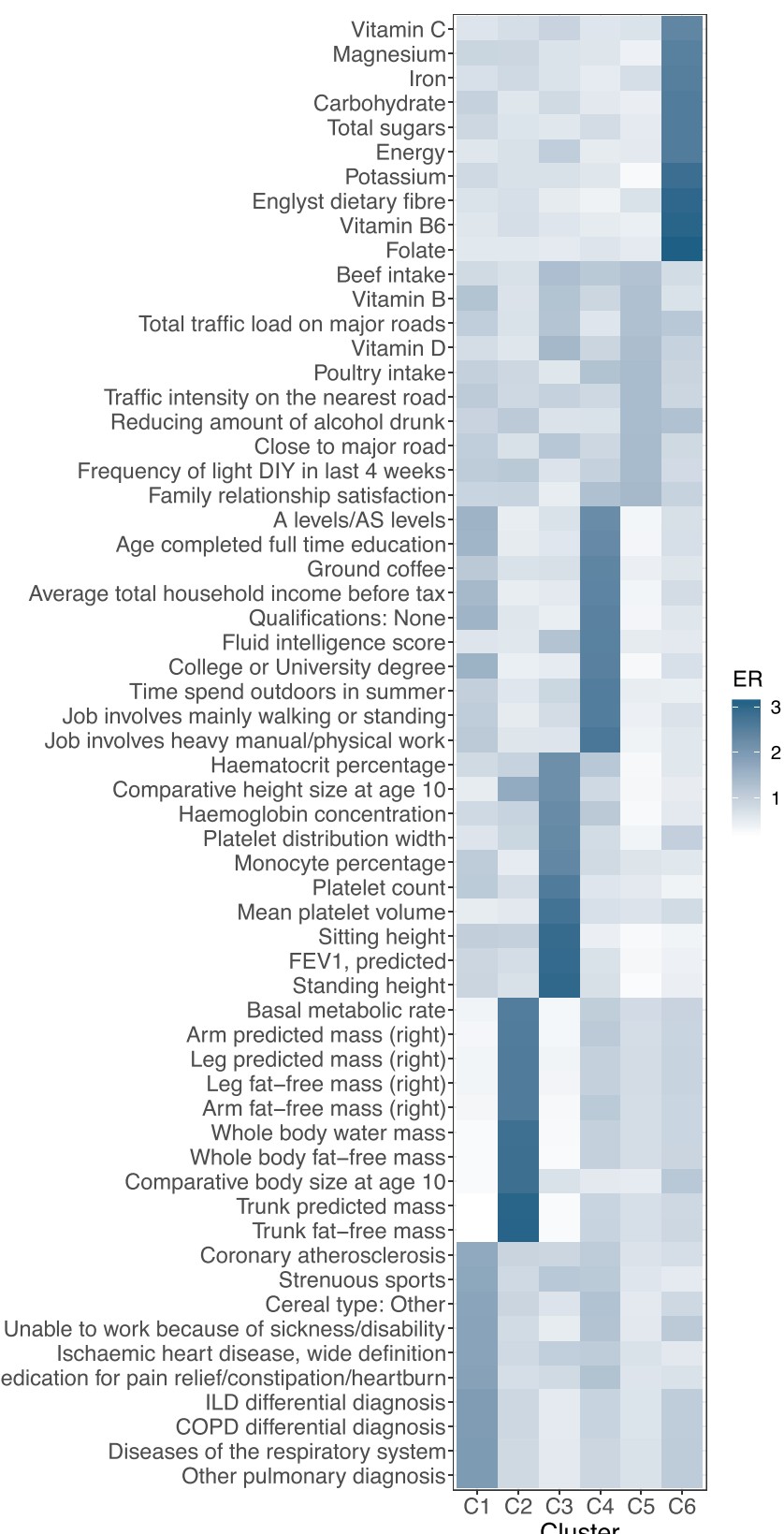

**Fig. 3 | Heatmap of the enrichment ratio of the top 10 traits in each cluster.** K-means clustering of BMI revealed six clusters with the following trait enrichment ratios.

ran a stepwise MVMR on these 19 candidate confounder traits (Supplementary Data 9). During the creation of the Z-score matrix of SNPs and traits, only twelve traits had at least three genome-wide significant and independent SNPs whose effects could be used in the analysis,

leaving us with a total of 683 SNPs across these twelve traits and BMI. The twelve traits were: 'Time spent watching television (TV)', 'Usual walking pace', 'Past tobacco smoking', 'Cereal type: Muesli', 'Frequency of tiredness/lethargy in last 2 weeks', 'Frequency of depressed mood in

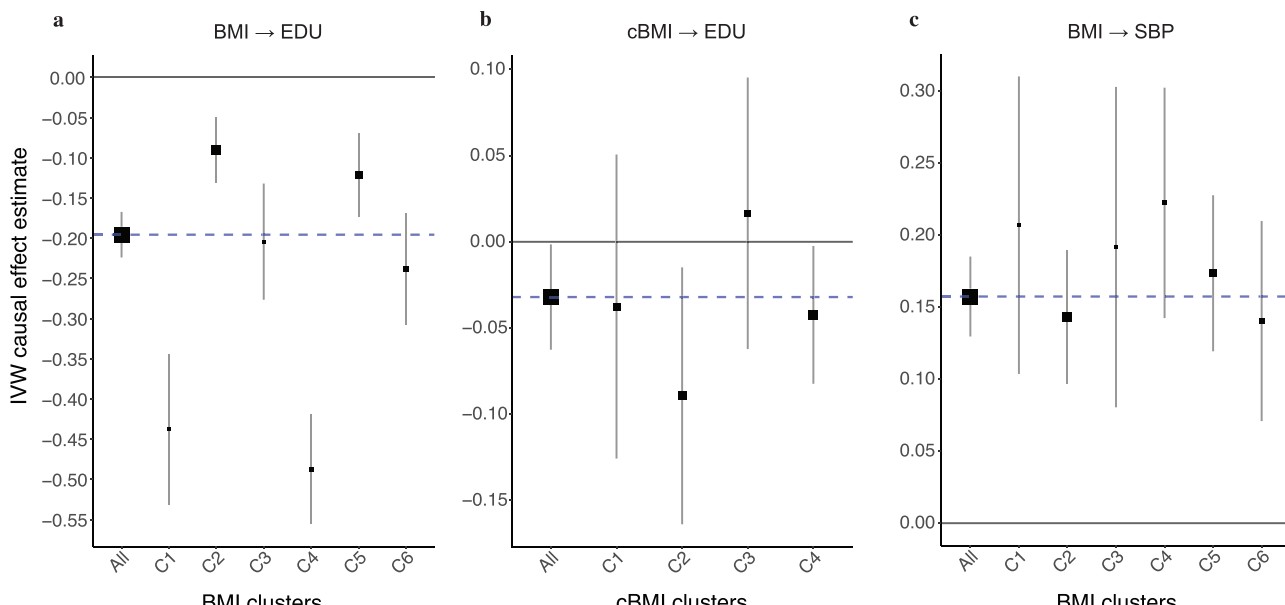

**Fig. 4 | Forest plot of IVW causal effect estimate on outcome using either all exposure IVs (All) or cluster-specific IVs (C1..C4/C6).** Panel **a** shows causal effect estimates of adult BMI on EDU, panel **b** proxy of childhood BMI (cBMI) on EDU, and panel **c** adult BMI on SBP. Vertical error bars represent the point estimate ± 1.96 × standard error (SE). The blue vertical line represents the causal effect estimated using all BMI/cBMI IVs. Box sizes of clusters represent the proportion of the number of IVs in each cluster to the total.

last 2 weeks', 'Public transport', 'Walking for pleasure', 'Weekly usage of mobile phone in last 3 months', 'Eating eggs, dairy, wheat, sugar', 'Symptoms, signs and abnormal clinical and laboratory findings', and 'Average weekly beer plus cider intake'. Of these, only the first four had a significant causal effect estimate on EDU (Bonferroni corrected *p*-value < 0.05/12) based on stepwise MVMR, and were subsequently used as exposures alongside BMI in a standard MVMR analysis.

To ensure the strength of the IVs used in the MVMR analysis, we calculated the conditional F-statistic and the MVMR causal effect estimate of BMI given various combinations of the four remaining candidate confounder traits. We saw the expected trend of a decreasing conditional F-statistic with the addition of traits and their IVs to the analysis (see Supplementary Fig. 7). We note that the causal effect estimate of BMI on EDU decreases when any combination of the candidate confounder traits is used with BMI as exposure in comparison to the univariable MR causal effect estimate of BMI on EDU (−0.19,

### Table 1 | MVMR analysis results of BMI and three candidate confounder traits on educational attainment

| Trait | Description | α estimate | SE | *P*-value |
|---|---|---|---|---|
| 1070 | Time spent watching television (TV) | -0.2771 | 0.0256 | 4.63E-25 |
| 1249 | Past tobacco smoking | 0.1592 | 0.0218 | 7.85E-13 |
| 1468_4 | Cereal type: Muesli | 0.2930 | 0.0383 | 7.96E-14 |
| 21001 | Body mass index (BMI) | -0.0455 | 0.0106 | 2.07E-05 |

α: causal effect estimate.

### Table 2 | Cross table of BMI IVs clustered using PWC-MR and MR-Clust

| MR-Clust<br>PWC-MR | Cluster1 | Cluster2 | Cluster3 | Cluster4 | Cluster5 | Cluster6 |
|---|---|---|---|---|---|---|
| **Cluster1** | 13 | 1 | 0 | 13 | 0 | 5 |
| **Cluster2** | 15 | 38 | 21 | 26 | 32 | 29 |
| **Null** | 4 | 59 | 14 | 2 | 37 | 15 |

*p*-value = 7.11 × 10$^{-41}$). We settled on the combination of candidate confounder traits yielding a conditional F-statistic for BMI of 10.19, for which the corresponding causal effect estimates are reported in Table 1 below. This choice was a compromise between two sources of biases: weak instrument bias *vs.* upward bias due to omitting relevant confounders.

### Comparison with MR-clust

Other known IV clustering methods include MR-Clust[21], which attempts to cluster variants with similar causal effect estimates together following the hypothesis that exposures can affect an outcome by distinct causal mechanisms to varying extents. MR-Clust also accounts for the possibility of spurious clusters by assigning IVs with uncertain causal effect estimates to 'null' or 'junk' clusters.

We compared the PWC-MR clustering of BMI IVs against that of MR-Clust with EDU as the outcome. The MR-Clust results revealed two main clusters as well as a 'null' cluster. Cluster #1 had 35 SNPs, 13 of which had an inclusion probability greater than 80%. Cluster #2 had 171 SNPs, 36 of which had an inclusion probability greater than 80%, and the remaining 142 SNPs were categorised into the 'null' cluster as seen in Supplementary Fig. 8. The mean causal effect estimate of SNPs in cluster #1 was −0.496, whereas it was −0.246 for cluster #2. Searching for trait associations for the SNPs in each of the clusters revealed that body measurement traits like 'Arm fat mass' or 'Body fat percentage' are associated to SNPs in both clusters, while SEP-related traits such as 'Fluid intelligence score' or 'Time spent watching television' were associated to more SNPs in cluster #1 than in cluster #2.

Comparing the SNP clustering between the PWC-MR method against that of MR-Clust in Table 2 below, we see that cluster #1 in MR-Clust, which seems to be more strongly enriched for SEP traits than cluster #2, has SNPs that were similarly clustered in clusters #1 and #4 using PWC-MR, matching their large negative causal effect of BMI on EDU. However, the same distinct comparison cannot be made for SNPs in cluster #2 of MR-Clust.

Of the 12 Fisher's exact tests performed to examine the contingency of SNPs in the two separate sets of clusters, only four tests revealed a significant association: SNPs in cluster #1 of MR-Clust were

significantly associated with SNPs in clusters #1, #2 (lean-mass traits), #4 (SEP-related traits) and #5 of the PWC-MR clustering.

Given the differences between the two methods (where PWC-MR performs informative clustering of IVs based on external data, and then measures the MR causal effect estimates per cluster, compared to MR-Clust that clusters IVs based on the magnitude of their MR causal effects) we see a more biologically meaningful separation of SNPs using PWC-MR, shedding light on the various mechanisms through which BMI can act on EDU.

### Comparison with tissue-specific colocalisation analysis

With the aim of finding supporting evidence for the k-means clustering and enrichment analysis, we ran a genetic colocalisation analysis on BMI IVs and two types of tissue: subcutaneous adipose and brain, the results of which can be found in Supplementary Data 12 and 13 respectively.

Running a set of Fisher's tests to compute the overlap between the membership of the SNPs in the six clusters and their tissue of colocalization did not reveal any association between clusters and tissues, as seen in Table 3.

## Discussion

We have developed a method that performs informative clustering of IVs by utilising their association with a large number of traits. Our use of PheWAS data to guide the clustering of IVs has revealed distinct mechanisms by which exposure effects could act on outcomes. For our exposure, BMI, six distinct clusters were obtained through optimal K-means clustering. These clusters had well-defined trait enrichments, with clusters matching SEP-related, substrate, and body measurement traits. Estimating individual causal effects of each cluster on EDU as an outcome revealed heterogeneous causal effect estimates which allowed us to further strengthen our suspicion that the MR estimate for the causal effect of BMI on EDU is upwardly biased when using population-based SNP effect size estimates, due to confounding.

We note from MR analysis run using within-sibling GWAS data[18] that the causal effect estimate between BMI and EDU is −0.05 (95% CI: −0.09, −0.01), which is smaller than the causal effect estimate seen using population-based GWAS data (−0.19, 95% CI: −0.22, −0.16). Investigating the various mechanisms or pathways through which BMI could have a causal effect estimate on EDU through trait-enrichment analysis has revealed notable causal effect estimates from two clusters: one with a strongly negative MR estimate, the trait enrichment of which reflects shared mechanisms with socio-economic factors, and another cluster with close to zero causal effect estimate enriched for lean-mass traits. MR has typically presented bias due to heterogeneous causal effects emerging via distinct pathways, and bias due to confounding of the instrument-outcome association as being separate mechanisms. Here, we have illustrated that a pheWAS-based clustering approach can classify instruments into clusters, some of which correspond to different pathways, while others include IVs that are primarily confounder-associated. Our results have two major implications: 1)

The lean-mass-related IV cluster indicated a more plausible, close to zero causal effect of BMI on EDU, 2) The SEP-related IVs leading to an apparent, sizeable negative effect of BMI on EDU, are likely over-estimating the true underlying causal effect.

In order to substantiate our findings, we performed several sensitivity analyses. First, sib-regression-based MR of BMI on EDU recapitulated the close-to-zero causal effect obtained for the body-mass-specific cluster of IVs. This indicates that many IVs for adult BMI (from population-based GWAS) represent indirect (parental/dynastic) effects associated with a rearing-related parental trait, and not primarily with offspring BMI. Second, replacing adult BMI with childhood BMI (much less associated with SEP) as exposure in the PWC-MR analysis confirmed a negligible causal effect estimate (−0.03, 95% CI: −0.06, 0), and the four emerging clusters showed homogeneous causal effect estimates indicating a relative lack of confounding or biasing effects. This comparison was supported by the growing evidence showing that genetic variants have varying effects on BMI or body size at different stages of life[22,23], and that the UK Biobank proxy trait 'Comparative body size at age 10' captures childhood BMI well[19]. Noteworthy is the fact that the childhood BMI proxy we use is a coarsened trait in comparison to true childhood BMI, and thus its genetic effects are underestimated (due to noise dilution). Since the estimated causal effect is a ratio of outcome and exposure effects, then if the denominator is underestimated, the MR effect is likely to be overestimated. We have supported this intuition with simulations in Supplementary Methods 1.2, and Supplementary Fig. 12. Of the four clusters, one was strongly enriched for body-measurement/fat-mass traits whereas the second most strongly enriched cluster had only two mildly enriched SEP-related traits. This finding means that as opposed to adult BMI, childhood BMI genetics are unrelated to childhood (i.e. parental) SEP. Furthermore, out of the 41 adult BMI IVs that make up cluster #4 (SEP-related traits), only three were found to be in LD with childhood BMI IVs.

In Howe et al. (2022), assortative mating, dynastic effects, and population stratification were all considered candidate mechanisms for biased population-based GWAS effect estimates. Given our observations, a possible explanation is a dynastic effect of parental SEP traits acting as a confounder on the offspring's BMI and EDU in adulthood (as seen in Supplementary Fig. 9). This effect is direct on the offspring's adulthood EDU but could affect the offspring's adult BMI indirectly through either of two ways: (i) Parental SEP has a direct effect on the offspring's SEP as an adult, which in turn has an effect on offspring adult/late BMI[24], or (ii) parental SEP – as a determinant of childhood social circumstances – may have an effect through this on the offspring's adult BMI.

To explore the relevance of the obtained six clusters of IVs, we replaced EDU with SBP as the outcome of interest since within-sibling GWAS MR results showed no difference when compared to population GWAS MR results, indicating that there seems to be no bias in the causal effect estimate due to pleiotropy or confounding. Our analysis revealed that for the six clusters attributed to BMI, their causal effect estimate on SBP was homogeneous with the estimate using all SNPs (0.16, $p$-value = $1.09 \times 10^{-28}$). As there are no clear heterogeneous effects and the cluster causal effects agree, we can conclude that there are no major confounding effects biasing the causal effect estimate. It is reassuring to note that our PWC-MR approach does not always seek to identify distinct causal effects, confirming that confounding mechanisms are specific to certain exposure-outcome pairs.

Finally, our systematic confounder search coupled with stepwise MVMR has pinpointed TV watching, muesli eating, and past tobacco smoking as three candidate confounder traits that could bias standard MR analysis of the BMI-EDU relationship: upon accounting for these three traits, BMI exhibits a strongly attenuated causal effect on EDU, comparable to that of cluster #2 and the sib-regression MR estimate.

**Table 3 | Cross table indicating the number of genes whose expression colocalises in adipose/brain tissue with BMI**

|          | Cluster1 | Cluster2 | Cluster3 | Cluster4 | Cluster5 | Cluster6 |
|----------|----------|----------|----------|----------|----------|----------|
| **Adipose** | 9 | 9 | 14 | 3 | 6 | 5 |
| **Brain** | 3 | 3 | 4 | 1 | 2 | 4 |
| **Both** | 1 | 2 | 1 | 1 | 4 | 4 |
| **Neither** | 29 | 77 | 36 | 23 | 53 | 47 |

The colocalisation exercise was performed at loci-defined BMI IVs falling into particular clusters. Colocalisation was defined as the posterior probability of both GWAS and eQTL being associated is ≥ 0.8 in either brain or adipose tissue or both.

We acknowledge the fact that past tobacco smoking is unlikely to have an effect on EDU retroactively, similar to TV watching and other later-in-life traits, which we all consider to be acting as confounder-proxies or correlates of parental SEP. We have explored this further in Supplementary Methods 1.3 by introducing 'Smoking Initiation' into the candidate confounder traits, the results of which are found in Supplementary Table 1.

Comparing our method to other IV clustering methods such as MR-Clust does not reveal strong concordance in the findings. MR-Clust takes as input the association effects of the exposure and outcome as well as their association standard errors and attempts to cluster the exposure IVs based on the possible similarity between each IV's causal effect on the outcome. When using BMI and EDU as exposure and outcome respectively, MR-Clust revealed two main clusters alongside a null cluster. Both of the clusters were enriched for a variety of traits including body-measurement traits, both lean- and fat-mass, as well as SEP-related traits. The causal effect estimates of both clusters were strongly negative, similar to using all IVs in an MR analysis for this trait pair. The most apparent difference between the clustering of our method and that of MR-Clust is our use of external information (PheWAS data of the exposure IVs and various other traits) to reveal possible pathways and mechanisms through which the exposure manifests, independently of any outcome. MR-Clust, on the other hand, clusters the individual MR causal effects of IVs on a specific outcome based on their magnitude.

Another clustering method by Grant et al.[25] uses genetic variant associations with a set of traits to identify groups of IVs with similar biological mechanisms. Their method, NAvMix, uses a directional clustering algorithm and includes a noise-cluster to increase robustness to outliers. NAvMIX is demonstrated on BMI IVs and their associations to nine lifestyle or cardio-metabolic traits that have been previously shown to be related to BMI. Their results revealed 5 distinct clusters where they were able to identify a metabolically healthy obesity cluster that also had a small MR causal effect on coronary heart disease (CHD). However, we were unable to run their method using our data due to convergence issues arising when the number of traits used for PheWAS association increases. This comparison also highlights that the traits we include in the pheWAS analysis (and the subsequent clustering) have an important role in determining which biological mechanisms we can detect. For example, our analysis did not pick up the metabolically healthy obesity cluster, possibly because waist-to-hip ratio and other subcutaneous-vs-visceral fat proxy-traits were not included among the 407 selected phenotypes due to our filtering on genetic correlation with BMI ($r_G$ < 0.75). Without such filtering, PWC-MR reveals 5 clusters with significantly heterogeneous causal effects on EDU. These five clusters are very similar to the original six, with the original cluster #1 getting diffused into the other clusters. Reassuringly, the cluster that is strongly enriched for SEP-related traits has a large negative causal effect estimate of −0.53 (95% CI: −0.59, −0.48), whereas the cluster that is most enriched for body-measurement/fat-mass traits still had a much attenuated causal effect of −0.10 (95% CI: −0.14, −0.06).

Furthermore, we attempted to consolidate our findings of the k-means clustering and enrichment analysis by running a genetic colocalisation analysis on the 324 clustered BMI IVs and both subcutaneous adipose and brain tissue. However, we do not find an association between the cluster memberships of the IVs and their signal colocalization in brain or adipose tissue, possibly due to high false negative rates of colocalization combined with low eQTL sample sizes.

Our method has its own set of limitations: first, we are limited by the availability of traits with PheWAS data to support our informative clustering of IVs. This may lead to a failure in identifying key pathways and thus missing clusters representing important subgroups (mediator/sub-phenotype/confounder). Second, although it is not the most ideal handling of data, our binary traits are treated as continuous ones in our analysis. In large samples, linear and logistic regression effect estimates correlate very strongly and hence, it is likely that this choice did not impact the clustering[26]. Third, although we have attempted to minimise the arbitrary choice of parameters in our analysis, the genetic correlation threshold that determines which traits are too similar to the exposure and outcome trait is arbitrarily set at 0.75 for BMI and EDU, and could be modified but the emerging clusters may change as a consequence. Similarly, some p-value thresholds and type I error rate control were set at 5%, which may be viewed as arbitrary. Fourth, the identified potential confounder traits used in the MVMR analysis may act as simple proxies for true confounders. For example, exposure to current tobacco smoking or TV watching can be highly (genetically) correlated to the same or a similar exposure during early life (or even proxy a parental trait), hence it is rather the earlier version of the exposure which is likely to be the true confounder. Note however, that the role of our proxy confounders was to see the remaining causal effect of BMI on EDU upon conditioning on them. Fifth, while for the BMI-EDU relationship we had several lines of evidence pinpointing cluster #2 as the one yielding the most likely correct causal effect estimate, in general, we might not be able to decide which cluster(s) provide biologically meaningful causal effect estimate(s) and which ones may be linked to confounders. Lastly, we acknowledge that there are several other tests[27] that could be used in place of a t-test when excluding SNPs more strongly associated to other traits than our exposure or different MR methods used in our systematic confounder search, however both of these were simple exclusion or pre-selection steps that have little impact on the outcome of the results.

To conclude, we found that the classical MR estimate based on population GWAS leads to an overestimation of the BMI-EDU causal effect and identified a lean-mass-specific subgroup of IVs (cluster #2) that, we believe, yields a more reliable causal effect estimate. Still, we are uncertain whether this effect is exactly zero, or is just strongly attenuated. Our analysis also revealed that the unrealistically large standard MR estimate was driven by IVs that likely violate the pleiotropy assumption through being also linked to SEP. The attenuated estimate provided by our PWC-MR approach (cluster #2) is compatible with both the estimate based on sib-regression summary statistics (P-value of difference = 0.16) and the MVMR estimate (P-value of difference = 0.48), all of which are based on adulthood phenotypes. However, the estimate obtained for childhood BMI is slightly more attenuated than that of the PWC-MR method (P-value of difference = 0.024).

Equipping the MR toolkit with a range of different analytical strategies is critical for improving insights into epidemiological questions, and PWC-MR offers a number of attractive features: (i) it does not require summary statistics from within-family GWAS, which are typically scarce and available in much smaller samples and for a limited set of phenotypes (ii) it does not rely on association data from chronologically-correct exposures, which may face similar limitations as within-family GWAS (iii) in contrast to MVMR, which estimates a single causal effect, PWC-MR provides multiple causal effect estimates, some of which may reflect confounder effects, and others heterogeneous mechanisms of action, overall revealing biological insight that can be used in follow-up research.

## Methods
### Instrumental variable selection and PheWAS
As our primary analysis, we aimed to investigate the potential pleiotropy-patterns emerging from the grouping of IVs that are strongly associated with an exposure of interest, as outlined in Fig. 2a. With BMI selected as the exposure trait, we obtained IVs from the Neale group's UK Biobank GWAS analysis[28] (data sources can be found in Supplementary Data 1) by filtering for genome-wide significant SNPs (i.e. association p-value less than $5 \times 10^{-8}$) followed by linkage disequilibrium (LD)-based clumping using the TwoSampleMR R package[29] with the following parameters: $clump\_kb = 10,000$, $clump\_r2 = 0.001$, $pop = "EUR"$ to obtain independent IVs.

This left us with 348 BMI-associated IVs, for which we ran PheWASs across 1480 traits from the Neale group UK Biobank GWAS analysis[28]. We extracted for each trait and for each SNP the association effect and the corresponding standard error (SE), creating a data matrix of 348 SNPs by 1480 traits. For the 1480 traits, we also extracted details such as variable type, origin and complete sample size, among others.

**Quality control.** We removed traits from the PheWAS data matrix that had missing association effects as well as duplicates (keeping the most recent version). Furthermore, we filtered out traits for which the effective sample size was less than 50,000 due to their low power of association, leaving us with 424 traits.

Using genetic correlation data from the Neale group[28], we further removed traits that had a high genetic correlation with BMI, i.e. the exposure, ($r_G > 0.75$), to avoid obvious repetitions of traits closely related to it. The resulting association effect data matrix of 348 SNPs and 407 traits was then standardised (SNP effects are on a SD/SD scale) and used for further analysis. Note that for simplicity, effect sizes for binary traits were treated as those of continuous traits.

In order to test for invalid IVs, we performed a trait-wide variant of Steiger-filtering[30]. Specifically, for each SNP, we tested if any of the traits had a significantly stronger (in terms of explained variance) association compared to that of the exposure. The significance threshold for this one-sided t-test was corrected for using the total number of traits remaining ($p$-value $< 0.05/407$). This revealed 24 SNPs more strongly associated to traits other than BMI (such as 'Whole body water mass', 'Basal metabolic rate' and 'Sitting height') that were then removed from further analysis.

### K-means clustering and trait identification

With the aim of discovering distinct meaningful groups of SNPs among the 324 IVs, we proceeded with the clustering of the SNPxTrait association effect matrix using the K-means algorithm[31]. Taking the absolute standardised effects matrix, we normalised the data frame with respect to the SNPs, such that the variance of the SNP effects across all the traits equalled 1. We used the absolute effects to cluster, in order to ensure that negatively correlated traits were considered similar by the Euclidean distance-based similarity measure of the k-means clustering. We then compared the performance of the clustering with different number of clusters ranging from two to 50, by measuring the Akaike Information Criterion (AIC) score (for further model selection criteria, see Supplementary Methods 1.1, including Supplementary Figs. 10 and 11). After finding the number of clusters with the lowest AIC score (six clusters), we proceeded with the assignment of each SNP to one of the six clusters.

In order to identify traits that were particularly associated to SNPs in each of the six clusters, we computed an enrichment ratio (ER) in the following way:

For each trait $t$, we calculated the per-SNP average squared effect in a given cluster $j$, denoted as $\sigma_{j,t}^2$. Given that SNP $i$ belongs to cluster $j$, $\sigma_{j,t}^2$ was calculated as follows:

$$\sigma_{j,t}^2 = \frac{1}{|c_j|} \sum_{i \in c_j} \beta_{i,t}^2 \tag{1}$$

where $\beta_{i,t}^2$ represents the squared standardised effect of SNP $i$ on trait $t$ (not normalised across traits), $c_j$ represents the set of SNPs in cluster $j$ and $|c_j|$ its cardinality. We then normalised these per-SNP average squared effects for each cluster relative to the total effect across all clusters ($K$) to obtain the enrichment ratio (ER), $R_{j,t}$:

$$R_{j,t} = \frac{\sigma_{j,t}^2}{\frac{1}{K}\sum_{k=1}^{K} \sigma_{k,t}^2} \tag{2}$$

where $K$ is the total number of clusters. For each cluster ($j$), traits were then prioritised by the (highest) value of ER ($R_{j,t}$).

**Causal effect estimate per cluster.** We measured the cluster-specific IVW causal effect estimate on the outcome (EDU) using the standardised SNP effects in each cluster, and then compared these estimates to the causal effect estimate using all SNPs. We used the TwoSampleMR R package[29] for this analysis, and although we use two-sample MR techniques despite having a close to complete sample overlap, this does not lead to substantial biases[32]. Measures of IV heterogeneity were calculated using the Cochran's Q-statistic[33] for the IVW method for each cluster. Furthermore, average cluster-heterogeneity (per-IV variance) was also calculated for each cluster from the above-mentioned parameter.

As sensitivity analyses, PWC-MR was repeated twice, once with a different exposure trait (replacing BMI with childhood BMI), and another with a different outcome trait (replacing EDU with systolic blood pressure).

### Systematic confounder search

In order to decide which of the emerging clusters represents genetic confounding or true biological heterogeneity, we systematically searched for BMI-EDU confounders. To do this, we investigated the bidirectional causal effects that each trait had on both the exposure and the chosen outcome.

Firstly, an extra filtering step was done where traits that were highly genetically correlated with the outcome ($r_G > 0.75$) were removed from the total 407 traits of the previous analysis.

Then, we ran a bidirectional MR for the remaining traits using the TwoSampleMR R package[29], and obtained four sets of causal effect measurements per trait (bidirectional, two different outcome traits - BMI and EDU). To select bidirectional causal effect estimates from those calculated by the different methods in the TwoSampleMR package[29] (Weighted median, Inverse variance weighted, Simple mode, and Weighted mode), we ordered the p-values of the causal effect estimates for the four different methods and selected the estimate of the second most significant method to ensure that at least one other method supports the causal claim.

The next step was to identify the direction of causality. To do so, we performed a one-sided t-test to compare the strengths of the estimated causal effects between the trait and the exposure, BMI. More precisely,

$$t_{A,B} : = \frac{|\widehat{\alpha}_{A \to B}| - |\widehat{\alpha}_{B \to A}|}{\sqrt{SE_{A \to B}^2 + SE_{B \to A}^2}} \tag{3}$$

where $A$ and $B$ denote the examined traits, $\widehat{\alpha}_{A \to B}$ the causal effect estimate from $A$ on $B$ and $SE_{A \to B}$ the corresponding standard error (SE). The one-sided $P$-value is then calculated as $P = \Phi(t_{A,B})$: if $P < 0.05$ the $B \to A$ causal effect is nominally significantly larger, while if $P > 0.95$, the $A \to B$ direction is dominant. For all the $p$-values in between, it was challenging to assign a direction in which the causal effect was stronger, and thus these traits were not further categorised. The $p$-value thresholds we apply are not intended to suggest that there is a transition point at which the meaning of associations change, rather we use these as a heuristic that is required to control the type I error rate at an arbitrary (5%) threshold. We further tested varying one-sided $p$-value thresholds of more stringent ($P < 0.01$, $P > 0.99$) and more lenient nature ($P < 0.1$, $P > 0.9$), the results of which are found in Supplementary Data 10 and 11.

The same procedure was repeated to explore the relationship between the traits and the outcome trait (EDU). This allowed us to classify traits into candidate confounders, mediators, colliders and other categories (as seen in the middle panel of Fig. 2b). For example, a confounder was defined as a trait with a significantly larger effect on both exposure and outcome than the reverse. We then focused only on the confounders which can distort MR estimates, and filtered them further to make sure that they have at least a nominally significant MR causal estimate ($p$-value $< 0.05$) on both BMI and EDU. We were lenient

in our categorisation of candidate confounder traits as adding potentially irrelevant traits would not bias the multivariable causal effect of BMI in the next step. As our aim was not to reduce the total causal effect to the unmediated part (possible by including mediators in an MVMR) but to correctly estimate it, mediators were not considered further. Similarly, the inclusion of colliders into an MVMR does not alter the exposure's causal effect as previously seen[34], thus they too were not considered further. The same holds for traits with a direct effect on either the exposure or the outcome only.

Furthermore, to test how compatible the two lines of analysis were, we examined the cluster-specific enrichment ratio values for the set of candidate confounder traits we obtained.

**Multivariable MR**. Focusing on the candidate confounder traits resulting from the systematic search that could bias the causal effect estimate between the exposure-outcome pair, we first ran a stepwise multivariable MR (MVMR) (adapted from the bGWAS R package[35]) with them as exposures to test their effect on our chosen outcome, EDU.

To do this, we created a Z-score matrix combining all genome-wide significant SNPs ($p$-value less than $5 \times 10^{-8}$) and their Z-scores for each of the 19 candidate confounder traits and BMI, such that each SNP had an effect that is genome-wide significant for at least one of the candidate traits.

To obtain independent SNPs, we performed rank-based clumping. For this, we first ranked the absolute Z-scores across all SNPs for each trait (in descending order), and then for each SNP we obtained the highest (best) rank across traits, which was used as an importance score during the clumping process (LD-clumped $clump\_kb = 5000$, $clump\_r2 = 0.01$). We then further filtered out traits that had less than three instruments remaining. Note that any SNPs that fall in the HLA region (6p21.3) were removed for being strongly associated with multiple immune-related traits.

Using this Z-score matrix without our primary exposure (BMI) as input for step-wise MVMR, we obtained a final list of candidate confounder traits with significant multivariable causal effects ($p$-value < 0.05/12) on our chosen outcome (EDU).

Then, to minimise weak instrument bias when running MVMR, we calculated the conditional F-statistic for our primary exposure (BMI) given each of the surviving traits and their different combinations. Finally we ran standard MVMR using the combination of traits that produced a conditional F-statistic[36] $\geq 10$ (for BMI) alongside BMI, and examined the multivariable causal effect of BMI on EDU.

### Relation to other approaches

**Comparison with MR-Clust**. We compared the k-means clustering of BMI IVs against another IV clustering method called MR-Clust[21], which requires as input the unstandardised SNP effects on both the exposure and the outcome, as well as the standard error of the SNP on each. To do so, we performed a Fisher's exact test to examine the frequency distribution of SNPs in each of the k-means clusters against the MR-Clust clusters.

**Comparison with tissue-specific colocalisation analysis**. To further interpret the findings of the IV clustering, we sought to test if specific patterns of colocalisation in different tissue types appear for the different IV clusters.

To do this, we reran the steps detailed in Leyden et al.[37] for the 324 BMI IVs used in this work. For each IV, we tested for genetic colocalisation between the BMI GWAS data and the gene expression (eQTL) data of both subcutaneous adipose and brain tissue (data sources can be found in Supplementary Data 1). For each SNP tested, we took a margin of 200kb up- and downstream, and used the coloc R package[38] to test the SNP's colocalisation with each gene found in that region, once using brain eQTL data, and another colocalisation using adipose eQTL data. We declared colocalisation if the posterior probability of

the model sharing a single causal variant was larger than 80%. For each of the aforementioned clusters, we investigated if the IVs were more strongly enriched for or depleted in one tissue or the other using Fisher's exact test.

### Reporting summary

Further information on research design is available in the Nature Portfolio Reporting Summary linked to this article.

## Data availability

The origin and unique identifier of each of the summary statistics data used is referenced in Supplementary Data 1. The UK Biobank summary statistics data used in this study can be downloaded from http://www.nealelab.is/uk-biobank. BMI meta-analyzed GWAS and adipose meta-analyzed cis-eQTL can be obtained from Leyden et al. [37] with permission, and the brain cis-eQTL data can be downloaded from https://yanglab.westlake.edu.cn/software/smr/#DataResource. All results and data generated during this study are included in this published article and its supplementary information files.

## Code availability

The source code[39] for this work can be found on https://github.com/LizaDarrous/PheWAS-cluster(https://doi.org/10.5281/zenodo.10466847).

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

## Acknowledgements
This research has been conducted using the UK Biobank Resource under Application Number 16389. Z.K. was funded by the Swiss National Science Foundation (310030_189147 and 32003B_173092). G.H. is funded by the Wellcome Trust and Royal Society [208806/Z/17/Z]. G.H. and G.D.S. work within the MRC Integrative Epidemiology Unit at the University of Bristol (MC_UU_00032/01). For computations, we used the CHUV HPC cluster. We are grateful for the useful discussions on MVMR with Eleanor Sanderson.

## Author contributions
L.D. and Z.K. conceived and designed the project. Z.K. supervised all statistical analyses. L.D. implemented the research and performed the analyses. L.D. and Z.K. prepared the first draft of the manuscript. L.D., Z.K., G.H. and G.D.S. contributed to the review and editing of the manuscript.

## Competing interests
The authors declare no competing interests.
