## [Peer Review File · Nature Communications]

PheWAS-based clustering of Mendelian Randomisation instruments reveals distinct mechanism-specific causal effects between obesity and educational attainmentReviewer #1 (Remarks to the Author):

Many thanks for the opportunity to revise this interesting paper focusing on clustering possible instruments for MR analyses according to their relationships with hundreds of traits. It provides novel insights into the nature of existing causal relationships, in particular findings (like BMI -> education) that appear spurious on theoretical grounds. I have a few comments, detailed below.

Can the authors explicitly address whether they think that the causal relationship from BMI to education should be 0. With their method, the authors find that there is a residual effect which is still half the initial effect, quite significant, and higher than estimates from all the other sensitivity methods they implement. One reading of this set of result is that the proposed method is not as able to remove bias as efficiently as other available methods.

The fit in K-Means clustering can be assessed by several indices. Why choose AIC only and not BIC for example? Typically, several of these indices are presented and a consensus is found based on convergence and/or consideration of how meaningful groupings are for different solutions (e.g. 7 groups might fit slightly better but the 7th group might include very diverse sets of SNPs and does not add any meaningful content to other clusters). Choosing only one index may give the false impression of certainty in choosing the number of groups, when there is always a (fairly large) degree of arbitrary involved in such clustering methods.

Using squared effects of SNPs makes sense for continuous traits as this captures the variance explained in the trait by this particular SNP. But what about binary traits? Can this lead to bias, in particular if the % of binary traits differ in each cluster, which could affect enrichment statistics?

Please number equations.

This is certainly a misunderstanding on my part, but for the enrichment statistics normalized across clusters, why not use J instead of K for total number of clusters, we seem to end up with both j and k representing a given cluster in that equation?

Prioritising by the highest value only could be problematic. We could have value of 0.6, 0.61 and 0.62 where difference would not be meaningful. Is there a test of whether there is real heterogeneity in enrichment across clusters?

Revise caption figure 1: why use U as genetic confounder, letter usually used for unobserved confounding, in particular when G is also used in figure?

"Mediators and colliders were not considered further since their inclusion into an 198 MVMR does not alter the exposure's causal effect" I assume this is based on theoretical considerations rather than an empirical test? Please further justify.

Lots of arbitrary thresholds involved in selecting confounders, can you please comment on how that can affect analyses? Any sensitivity analysis involving those thresholds to check whether ultimate causal estimates are sensitive to those choices.

What about overfitting: you searched a lot of traits for potential confounding effects, how many of the selected potential confounders result from overfitting? Any validation (even split sample?)

Further explain principles for MR-Clust and difference of current method with that one (i.e. why do we need another method?).

Functional significance of traits sometimes appears confusing (ie grouping platelet count and body height). What impact does this have on the validity of results and their interpretation?

Related to the previous question, how is the clustering method better than a more theoretically driven selection of curated instruments (i.e. comparing instruments known to be related to metabolism to have a less confounded instrument for BMI).

Reviewer #2 (Remarks to the Author):

This paper uses phewas-based cluster of genetic instruments to attempt to decompose causal effects measured by Mendelian randomization (MR) into components attributable to distinct biological pathways. The method described is applied to the relationship between BMI and educational attainment (EDU).

I like this idea. I think it is timely and a positive contribution to the MR literature. I have a few, mostly minor, comments about the execution of the method and the writing below.

1. In the K-means clustering, I am not sure if the enrichment ratio is computed using effect estimates after normalizing effects for each variant across traits or before. I think this could have some effect on the ratios and it would be nice to have this explained and a sentence or so justifying the choice. My instinct would be to use the estimates pre-normalization but there may be reasons to prefer the other way.

2. I think the DAG in Figure 1a is a bit convoluted and doesn't help with understanding the logic. It also uses notation (X_{early} , X_{late} etc) that doesn't appear elsewhere in the paper. I think this DAG is too focused on the specific case that some of the heritable confounding is mediated by parental nurturing effects. It would be easier to understand if the DAG used the same notation as used elsewhere and demonstrated the relationship of the variant clusters with the traits in the DAG. (Having said that, I do like the figure in 1b which is very clear to me).

3. In section 2.3, the one-sided t-test for orienting the causal effects is described a bit vaguely. It would help to explain what numbers go in and which are being compared. The sentence reads "To do so, we performed a one-sided t-test on the estimated causal effect between the trait and the exposure, BMI." However, inferring from context, I believe the t-test is comparing an estimate of the effect of exp. on BMI to an estimate of the effect of BMI on exp. I would suggest clarifying this paragraph and including some supplementary text giving the explicit procedure/calculations used.

4. I was confused about the details of the MVMR analysis. In section 2.3.1, it is described that you finally have a matrix of 274 SNPs for 5 traits. It's not clear to me if this includes BMI (in which case I would expect more SNPs since BMI alone had 300+ instruments) or not. Additionally, in section 3.3.4 describing the results of this, you say that there are 19 candidate confounders. Is it a typo that 2.3.1 says 5 rather than 19 or are these referring to different sets of traits?

5. In Table 1, it seems like you find that increased "past tobacco smoking" has a positive effect on years of education. One potential issue with this is that most of a person's smoking history occurs after the end of the time that one is receiving education, so this appears to be an effect backwards in time. Even if interpreted as "genetic predisposition to smoke" this effect seems a bit unexpected as having any major impact either one way or the other on educational attainment. Can you comment on this relationship at all?

6. In Section 3.3.2 you say "Thus, we obtained a massively attenuated causal effect of BMI on EDU, when childhood BMI is used as an exposure." I am wondering if it makes sense to directly compare the magnitude of the effect estimates for adult and childhood BMI, given that childhood BMI was proxied by a three level categorical variable of relative body size. Is a one standard deviation increase in this categorical variable really comparable to a one standard deviation increase in BMI? (If BMI were

approximately normally distributed, two individuals could differ by 1 SD of BMI but still both be in the middle third of the distribution and therefore have the same level of "relative body size").

7. It would be good to add labels to Table 2 for K-means and MR-Clust. It would also be nice to have a little of the interpretation of this analysis in the results section rather than in the discussion, since it is not immediately clear from methods or results what the point of this analysis is.

REVIEWER COMMENTS & RESPONSES - NCOMMS-23-11823

Firstly, we would like to thank both reviewers and the editor for considering our research, and for reading our findings. Secondly, we are grateful for the insightful and constructive comments of all reviewers. They have led to significant improvements and clarifications of the original work, and have helped us fortify our methods with more stringent testing and analysis performed and detailed below.

Reviewer #1 (Remarks to the Author):

Many thanks for the opportunity to revise this interesting paper focusing on clustering possible instruments for MR analyses according to their relationships with hundreds of traits. It provides novel insights into the nature of existing causal relationships, in particular findings (like BMI → education) that appear spurious on theoretical grounds. I have a few comments, detailed below.

1. Can the authors explicitly address whether they think that the causal relationship from BMI to education should be 0. With their method, the authors find that there is a residual effect which is still half the initial effect, quite significant, and higher than estimates from all the other sensitivity methods they implement. One reading of this set of result is that the proposed method is not as able to remove bias as efficiently as other available methods.

Thank you for your comment, indeed this was under-discussed in the previous version of the manuscript. We modified the text to address this excellent point:

We have added to other lines of evidence that the classical MR estimate based on population cohort effect size estimates leads to a substantial overestimation of the BMI→EDU causal effect, though whether the effect is zero remains uncertain. Arguments for a zero causal effect can be that (i) the exposure (adult BMI) occurs after the establishment of educational attainment, hence strictly speaking it cannot have a retrospective causal effect; (ii) if we use (a proxy for) pre-educational BMI as exposure, then the causal effect is compatible with zero. Other approaches, such as sib-regression based causal effect estimate and MVMR estimate, point to an attenuated causal effect whose confidence intervals do not overlap the null, thus either a small causal effect is real, or even these approaches fail to completely remove all potential MR violations. The attenuated estimate provided by our PWC-MR approach is compatible with both the estimate based on sib-regression summary statistics (P-values difference = 0.16) and the MVMR estimate (p-diff = 0.48), all of which are based on adulthood phenotypes. However, the estimate obtained for childhood BMI is slightly more attenuated than the PWC-MR method (p-diff 0.024).

Equipping the MR toolkit with a range of different analytical strategies is critical for improving insights into epidemiological questions, and PWC-MR offers a number of features that compliment other approaches: (i) It does not require summary statistics from within-family GWAS, which is typically scarce and available in much smaller samples and for a limited set of phenotypes. (ii) It does not require association data from an early exposure (which suffers from the same problems as within-family GWAS). (iii) PWC-MR provides multiple causal effect estimates, some of which may reflect confounder effects, but others may reflect genuinely heterogeneous mechanisms of action.

2. The fit in K-Means clustering can be assessed by several indices. Why choose AIC only and not BIC for example? Typically, several of these indices are presented and a consensus is found based on convergence and/or consideration of how meaningful groupings are for different solutions (e.g. 7 groups might fit slightly better but the 7th group might include very diverse sets of SNPs and does not add any meaningful content to other clusters). Choosing only one index may give the false impression of certainty in choosing the number of groups, when there is always a (fairly large) degree of arbitrary involved in such clustering methods.

Indeed, it is always a good idea to have multiple concurrent indices of clustering performance, however the clustering we perform has a peculiar aspect that renders BIC less applicable. BIC introduces a stronger penalty term, $k \cdot \log(n)$, where k is the number of clusters and n corresponds to the number of (independent) samples used. However, in our case n represents the number of traits, which are highly correlated. Also, the more traits are used to cluster the SNPs, the more clusters we expect to obtain as they allow for a more fine-grain resolution of the underlying biological mechanisms. For these reasons, we do not believe that BIC is an appropriate measure to quantify clustering fit in this situation.

Still, to answer the reviewer's question, using BIC, we end up with 2 clusters being the optimal for BMI SNPs with heterogeneous causal effects on EDU (cluster 1 = -0.13 (6.61E-16), cluster 2 = -0.34 (6.15E-44)), and their enrichment reflects a clear distinction between enrichment for lean-mass and body related traits in cluster 1 and a mixed bag of trait enrichment for cluster 2 including lung/height/blood and SES-proxy traits. Therefore, the BIC-based do not alter the main message and only leads to coarser grain clusters.

We also tried forcibly increasing the number of clusters to 8 in the hopes of achieving more distinction in enrichment. We observed similar heterogeneous causal effects on EDU, where the smallest and largest effects were from clusters enriched for lean mass and SES-related traits respectively. As for the rest of the clusters, another 2 were strongly enriched for food supplements and a mix of height/blood/lung measurement traits, another was enriched for a mix of diseases and three other clusters had low enrichments for miscellaneous traits.

IVW causal effect estimates – K-means clustering based on BIC

IVW causal effect estimates – K-means clustering based on AIC

Figure 1 - Left: Causal effect estimates of BIC-clustered BMI SNPs on educational attainment
Right: Causal effect estimates of clustered (forced 8 clusters) BMI SNPs on educational attainment

3. Using squared effects of SNPs makes sense for continuous traits as this captures the variance explained in the trait by this particular SNP. But what about binary traits? Can this lead to bias, in particular if the % of binary traits differ in each cluster, which could affect enrichment statistics?

The trait-enrichment was calculated in a trait-to-trait basis as follows: we looked at the average squared effects of SNPs in each cluster on a focal trait and finally compared these average effects across clusters. Thus, this enrichment measure is invariant to effect rescaling, hence liability scale transformed effects would give identical results.

4. Please number equations.

Our apologies, we have now numbered all the equations in the manuscript.

5. This is certainly a misunderstanding on my part, but for the enrichment statistics normalized across clusters, why not use J instead of K for total number of clusters, we seem to end up with both j and k representing a given cluster in that equation?

Thank you for pointing this out, we have reworded the sentence in line 151 to be clearer:

...where $\beta_{i,t}^2$ represents the squared standardised effect of SNP i on trait t , c_j represents the set of SNPs in cluster j and $|c_j|$ its cardinality. We then normalised these per-SNP average squared effects for each cluster relative to the total effect across all clusters (K) to obtain the enrichment ratio (ER)...

Furthermore, in Eq (2) we cannot use j as a running index in the denominator, since j is already used in the numerator to index a specific cluster. For this reason, we chose another letter (k) for the running index.

6. Prioritising by the highest value only could be problematic. We could have value of 0.6, 0.61 and 0.62 where difference would not be meaningful. Is there a test of whether there is real heterogeneity in enrichment across clusters?

This is a good point, and we agree that a rigorous quantification of the enrichment was missing. We have now performed a statistical test to see for each trait which cluster(s) have significantly higher heritability than the rest of the clusters. These tests provided very strong supportive evidence that some of the top ten enriched traits listed for each cluster are the same traits (top ten shown in figure below) that are very significantly deviating from the null expectation.

Top10Traits_Cluster1	AnovaP_Cluster1	Top10Traits_Cluster2	AnovaP_Cluster2	Top10Traits_Cluster3	AnovaP_Cluster3
None: Medication for pain relief, constipation, heartburn	9.1E-09	Trunk fat-free mass	7.6E-22	Forced expiratory volume in 1-second (FEV1), predicted	8.9E-22
Other pulmonary diagnosis	3.3E-07	Trunk predicted mass	8.2E-22	Standing height	1.5E-21
Diseases of the respiratory system	3.3E-07	Whole body fat-free mass	1.7E-19	Sitting height	2.2E-13
COPD differential diagnosis	7.7E-07	Whole body water mass	2.8E-19	Reticulocyte count	4.5E-11
ILD differential diagnosis	7.7E-07	Arm fat-free mass (left)	3.9E-19	Reticulocyte percentage	1.3E-10
Unable to work because of sickness or disability	8.8E-06	Arm predicted mass (left)	5.0E-19	High light scatter reticulocyte count	3.4E-09
Overall health rating	5.4E-05	Arm fat-free mass (right)	2.6E-18	High light scatter reticulocyte percentage	1.4E-08
Ischaemic heart disease, wide definition	5.9E-05	Arm predicted mass (right)	4.0E-18	Platelet count	1.2E-07
Smoking status: Current	1.1E-04	Basal metabolic rate	1.1E-17	Forced vital capacity (FVC), Best measure	1.8E-07
Current tobacco smoking	1.2E-04	Leg fat-free mass (right)	2.6E-16	Mean platelet (thrombocyte) volume	2.2E-07
Top10Traits_Cluster4	AnovaP_Cluster4	Top10Traits_Cluster5	AnovaP_Cluster5	Top10Traits_Cluster6	AnovaP_Cluster6
Qualifications: None of the above	7.9E-22	Potassium	2.8E-05	Vitamin B6	1.0E-35
Age completed full time education	3.6E-20	Age completed full time education	3.9E-05	Folate	2.5E-35
Qualifications: College or University degree	3.1E-19	Qualifications: None of the above	5.3E-05	Englyst dietary fibre	2.1E-30
Job involves mainly walking or standing	3.6E-18	Reticulocyte count	1.1E-04	Potassium	1.1E-25
Qualifications: A levels/AS levels or equivalent	2.8E-16	Qualifications: College or University degree	1.3E-04	Iron	2.1E-22
Job involves heavy manual or physical work	5.6E-16	Qualifications: A levels/AS levels or equivalent	1.5E-04	Carbohydrate	1.2E-21
Average total household income before tax	1.8E-14	Standing height	1.7E-04	Energy	1.4E-20
Time spend outdoors in summer	3.5E-14	High light scatter reticulocyte count	1.8E-04	Total sugars	1.8E-20
Qualifications: CSEs or equivalent	3.1E-13	Reticulocyte percentage	3.0E-04	Vitamin C	1.0E-15
Time spent watching television (TV)	3.9E-12	Forced expiratory volume in 1-second (FEV1), predicted	3.8E-04	Magnesium	2.7E-15

Figure 2 - Table representation of the top 10 traits with significantly different heritability in each cluster compared to all the other clusters. Clusters 2 and 4 show significant heterogeneity in enrichment for traits related to lean mass and SEP respectively.

7. Revise caption figure 1: why use U as genetic confounder, letter usually used for unobserved cofounding, in particular when G is also used in figure?

Thank you for noting the caption of figure 1a, as it was missing the definition of G, which is the genetic instrument used in MR. The genetic confounder is mostly unknown; hence U still refers to this fact. But to better emphasise that U is a genetic confounder, we switched to U_G .

8. "Mediators and colliders were not considered further since their inclusion into an 198 MVMR does not alter the exposure's causal effect" I assume this is based on theoretical considerations rather than an empirical test? Please further justify.

We have rephrased the justification why not to include colliders and mediators:

Including a mediator could change the multivariable effect estimate to only direct causal effect. However, we are not after the direct effect, since the total effect of such exposures on the outcome is still real, but it may be indirect. Our aim was not to reduce the causal effect to only the unmediated part, but to correctly estimate the total causal effect. Including colliders in the MVMR does not alter the estimate, which has been shown in detail in previous literature [e.g. <https://www.ncbi.nlm.nih.gov/pmc/articles/PMC6734942/>, where scenario #2 covers the collider case and scenario #4 the mediator case, results are in Supplementary Table 2].

9. Lots of arbitrary thresholds involved in selecting confounders, can you please comment on how that can affect analyses? Any sensitivity analysis involving those thresholds to check whether ultimate causal estimates are sensitive to those choices.

Indeed, these thresholds could have been differently chosen. For this reason, we provide results for milder and more stringent thresholds to demonstrate that the results are robust to such choices. We change the P value threshold for the difference between the bidirectional causal effect of the exposure and the candidate confounder as well as the outcome and the same candidate confounder to either 0.01 for a more stringent threshold or 0.1 for a more lenient one (P value less

than either indicates a stronger causal effect from the candidate confounder on the exposure or the outcome).

In our more lenient threshold attempt, we obtain 27 candidate confounder traits, 7 of which survive stepwise MVMR. Calculating the conditional F-statistic of BMI given the various combinations of these 7 traits, we obtain a value of 10.75 for the combination of 'Time spent watching television (TV)', 'Time spend outdoors in summer', 'Average weekly beer plus cider intake', where BMI's causal effect on EDU conditional on those three was also attenuated to -0.07 ($P = 2.61E-12$).

As for our more stringent threshold, we obtain 3 candidate confounders, where only one survives stepwise MVMR: 'Usual walking pace'. BMI's causal effect on EDU conditional on that trait is -0.05 ($P = 0.001$), however with a conditional F-statistic of 4.1. Both these results have now been added to Supplementary Tables 10 & 11.

Thus, we believe that a P value threshold of 0.05 (or even 0.1) is lenient enough to allow a reasonable amount of candidate confounder traits to be tested. The key is the following stepwise-MVMR and the conditional F-statistic calculation that keep the most suitable candidate confounder traits for the final MVMR analysis.

10. What about overfitting: you searched a lot of traits for potential confounding effects, how many of the selected potential confounders result from overfitting? Any validation (even split sample?)

We thank the reviewer for this question. While we indeed tested a large number of traits as potential confounders, they all have very strong ($P < 10^{-12}$) multivariable causal effects on education, which cannot noticeably suffer from Winner's curse or lead to overfitting. While some of these may not strictly speaking be confounders (as their effect on BMI may be less strong), their purpose is to simply check the remaining (multivariable) causal effect of BMI on EDU, while accounting for these factors. Indeed, if we had included traits with milder MR P-values for their effect on EDU, it would have led to overfitting. However, this is not the case here.

In addition, to minimise weak instrument bias, we calculated the conditional F-statistic of BMI given various combinations of the 4 surviving confounders. We find that the combination of three of them (Time spent watching television, Past tobacco smoking, Cereal type: Muesli) gives a conditional F-statistic of 10.19, respecting the common threshold of 10. We then proceed to run MVMR with these 3 confounders and BMI as exposures to measure their causal effect on the outcome conditional on each other.

It is important to note that the creation of the Z-matrix process has been amended to be fairer in the genome-wide significant IV selection and clumping process to all traits. The new clumped and filtered (all traits must have at least 3 GW significant SNPs) Z-matrix has 682 SNPs and their Z-score across all 12 candidate confounders and BMI. More details can be found in the updated section '2.3.1 Multivariable MR'.

11. Further explain principles for MR-Clust and difference of current method with that one (i.e. why do we need another method?).

We thank the reviewer for this comment. We have amended the relevant section in the discussion to highlight the differences between MR-Clust and PWC-MR:

The aim of MR-Clust and PWC-MR is very similar, but PWC-MR uses additional external data to decompose the different exposure pathways and then it looks at their effect on the outcome, while MR-Clust clusters individual causal effects based on their magnitude to come up with the clusters.

12. Functional significance of traits sometimes appears confusing (ie grouping platelet count and body height). What impact does this have on the validity of results and their interpretation?

Indeed, some of the clusters seem to be a mixed bag and increasing the number of clusters might split these groups of SNPs into separate clusters, making the functional annotations more homogeneous. Adding more traits could probably also help to obtain cleaner (and more) clusters. We focussed in our research on clear clusters such as C2 and C4, as other clusters have less clear interpretations, and acknowledge this as a subject for future research.

13. Related to the previous question, how is the clustering method better than a more theoretically driven selection of curated instruments (i.e. comparing instruments known to be related to metabolism to have a less confounded instrument for BMI).

We thank the reviewer for raising this interesting point. Manual curation of instruments would in theory be the ideal way to get a more accurate understanding of the pathways underlying a given exposure trait and its causal effects on other traits. However, with the increasing number of instruments it is becoming less and less feasible, hence we believe that more automation will be necessary in the future. Indeed, our approach is one of the first steps towards this goal. Others have tried hybrid (semi-manual) approaches for metabolically favourable BMI, which focussed on a very small set of traits and one specific subgroup of SNPs¹. Our approach is less manual, but provides a broader overview.

¹ Yaghootkar, Hanieh et al. "Genetic Evidence for a Link Between Favorable Adiposity and Lower Risk of Type 2 Diabetes, Hypertension, and Heart Disease." *Diabetes* vol. 65,8 (2016): 2448-60.
doi:10.2337/db15-1671

Reviewer #2 (Remarks to the Author):

This paper uses phewas-based cluster of genetic instruments to attempt to decompose causal effects measured by Mendelian randomization (MR) into components attributable to distinct biological pathways. The method described is applied to the relationship between BMI and educational attainment (EDU).

I like this idea. I think it is timely and a positive contribution to the MR literature. I have a few, mostly minor, comments about the execution of the method and the writing below.

1. In the K-means clustering, I am not sure if the enrichment ratio is computed using effect estimates after normalizing effects for each variant across traits or before. I think this could have some effect on the ratios and it would be nice to have this explained and a sentence or so justifying the choice. My instinct would be to use the estimates pre-normalization but there may be reasons to prefer the other way.

We thank the reviewer for this perceptive comment. Indeed, the enrichment ratio is first computed for each trait by calculating the per-snp average squared effects per cluster. These effects were standardised across SNPs, but not across traits, in the quality control step to ease comparison in later steps. Unlike clustering, where we are searching for similarities across traits, the enrichment analysis is done for each trait independently of the others, hence it is not impacted by our normalisation. We have clarified this in the Methods section.

2. I think the DAG in Figure 1a is a bit convoluted and doesn't help with understanding the logic. It also uses notation (X_{early} , X_{late} etc) that doesn't appear elsewhere in the paper. I think this DAG is too focused on the specific case that some of the heritable confounding is mediated by parental nurturing effects. It would be easier to understand if the DAG used the same notation as used elsewhere and demonstrated the relationship of the variant clusters with the traits in the DAG. (Having said that, I do like the figure in 1b which is very clear to me).

We agree with the reviewer and have revised Fig1a to be more general as a standalone figure (shown below), and have moved the current version to the supplement, which is referred to when we discuss the possible explanations of confounding which may be in play for the BMI-EDU example.

Figure 3 - Directed Acyclic Graph (DAG) illustrating the complex relationship between exposure and outcome. G_j represents genetic instrument j with an effect β_j on exposure X . Exposure X is associated with outcome Y through K possible pathways of mediation or confounding denoted through the various $X_1 \dots X_K$ elements. The associations between the main exposure and the various elements denoted by the π arrows purposely do not show directionality to allow for both mediators and confounders. The causal effects on outcome Y are denoted by $\alpha_1, \alpha_2, \dots, \alpha_K$.

3. In section 2.3, the one-sided t-test for orienting the causal effects is described a bit vaguely. It would help to explain what numbers go in and which are being compared. The sentence reads "To do so, we performed a one-sided t-test on the estimated causal effect between the trait and the exposure, BMI." However, inferring from context, I believe the t-test is comparing an estimate of the effect of exp. on BMI to an estimate of the effect of BMI on exp. I would suggest clarifying this paragraph and including some supplementary text giving the explicit procedure/calculations used.

Thank you for your astute comment, it is correct and we have now amended this section to read more clearly the steps taken:

The next step was to identify the direction of causality. To do so, we performed a one-sided t-test to compare the strengths of the estimated causal effects between the trait and the exposure, BMI. More precisely,

$$t_{A,B} = \frac{|\widehat{\alpha}_{A \rightarrow B} - \widehat{\alpha}_{B \rightarrow A}|}{\sqrt{SE_{A \rightarrow B}^2 + SE_{B \rightarrow A}^2}}$$

where A and B denote the examined traits, $\widehat{\alpha}_{A \rightarrow B}$ the causal effect estimate from A on B and $SE_{A \rightarrow B}$ the corresponding standard error.

The one-sided P -value is then calculated as $P = \Phi(t_{A,B})$: if $P < 0.05$ the $B \rightarrow A$ causal effect is nominally significantly larger, while if $P > 0.95$, the $A \rightarrow B$ direction is dominant. For all the p -values in between, it was challenging to assign a direction in which the causal effect was stronger, and thus these traits were not further categorised.

4. I was confused about the details of the MVMR analysis. In section 2.3.1, it is described that you finally have a matrix of 274 SNPs for 5 traits. It's not clear to me if this includes BMI (in which case I would expect more SNPs since BMI alone had 300+ instruments) or not. Additionally, in section 3.3.4 describing the results of this, you say that there are 19 candidate confounders. Is it a typo that 2.3.1 says 5 rather than 19 or are these referring to different sets of traits?

Indeed, this was not clearly presented in the previous version of the manuscript. We have added panel (c) to Fig1 [previously Figure S1], which better explains the process: We scan through all 408 traits to select potential confounders of both BMI and EDU (bidirectional MR at a nominally significant level). This led us to 19 candidate confounders, which were subjected to stepwise MVMR to select a set of confounders with independent contribution to EDU, while maintaining a decent ($F > 10$) conditional F-statistic for BMI as exposure. This led us to the finally selected 3 confounder traits which were subjected to MVMR with BMI included as an additional exposure.

In addition to the answer above, it is important to note that the creation of the Z-matrix process has been amended to be more fair in the genome-wide significant IV selection and clumping process to all traits. The new clumped and filtered (all traits must have at least 3 GW significant SNPs)

Z-matrix has 682 SNPs and their Z-score across all 12 candidate confounders and BMI. More details can be found in the updated section '2.3.1 Multivariable MR', quoted below:

Focusing on the candidate confounder traits resulting from the systematic search that could bias the causal effect estimate between the exposure-outcome pair, we first ran a stepwise multivariable MR (MVMR) (adapted from the bGWAS R package) with them as exposures to test their effect on our chosen outcome, EDU.

To do this, we created a Z-score matrix combining genome-wide significant SNPs (p -value less than 5×10^{-8}) and their Z-scores for each of the 19 candidate confounder traits and BMI, such that each SNP had an effect that is genome-wide significant for at least one of the candidate traits.

To obtain independent SNPs, we performed rank-based clumping, where we first ranked the Z-scores of the SNPs for each trait (in descending order), and then selected the smallest ranking across traits to create a pseudo- p -value for all the SNPs. This pseudo p -value was normalised and was used alongside SNP location in the clumping process (LD-clumped $clump_kb = 5,000$, $clump_r2 = 0.01$) that kept 693 SNPs.

Note that any SNPs that fall in the HLA region (6p21.3) were removed for being strongly associated with multiple immune-related traits.

After clumping, we removed any trait that had less than three instruments, leaving us with a Z-score matrix of 683 SNPs across 13 traits.

Using this Z-score matrix without our principal exposure as input for step-wise MVMR, we obtained a final list of candidate confounder traits with significant multivariable causal effects (p -value $< 0.05/12$) on our chosen outcome.

Then, to ensure the strength of the instruments used for running MVMR, we calculated the conditional F-statistic for our main exposure (BMI) given each of the surviving traits and their different combinations.

Finally we ran standard MVMR using the combination of traits that produced a conditional F-statistic for BMI >10 , and BMI as exposures to estimate their conditional causal estimates on EDU.

5. In Table 1, it seems like you find that increased "past tobacco smoking" has a positive effect on years of education. One potential issue with this is that most of a person's smoking history occurs after the end of the time that one is receiving education, so this appears to be an effect backwards in time. Even if interpreted as "genetic predisposition to smoke" this effect seems a bit unexpected as having any major impact either one way or the other on educational attainment. Can you comment on this relationship at all?

We thank the review for this astute comment. Indeed, we agree that past tobacco smoking is unlikely to have a retroactive effect on education (or an effect at all, unlike education's effect on smoking). To further investigate this, we added smoking initiation (GWAS obtained from Saunders et al. 2022²), which on average occurs around the age of 17, to the MVMR analysis. We repeated first the stepwise-MVMR, obtained 'Smoking initiation', 'Time spent watching television (TV)', 'Cereal type: Muesli', and 'Usual walking pace' as candidate confounder traits with significant causal effects on EDU. Note that smoking initiation replaced past tobacco smoking in this step, as it no longer had a strong causal effect on EDU. Adding BMI to this set of exposures and then calculating its conditional F-statistic with their various combination, we discover that the

² Saunders, Gretchen R B et al. "Genetic diversity fuels gene discovery for tobacco and alcohol use." Nature vol. 612,7941 (2022): 720-724. doi:10.1038/s41586-022-05477-4

combination of the first three traits give a conditional F-statistic >10 and that BMI's conditional causal effect is severely attenuated:

Phenotype	Description	Estimate	SE	P	Conditional F-statistic
Smklnit	Smoking initiation	-0.1358	0.0122	7.66E-27	-
1070	Time spent watching television (TV)	-0.2617	0.0238	2.91E-26	-
1468_4	Cereal type: Muesli	0.2920	0.0341	5.39E-17	-
21001_irnt	Body mass index (BMI)	-0.0383	0.0103	2.01E-04	12.53

Smoking initiation, as seen, has a significantly negative causal effect on education, but we would like to iterate that it as well as the other candidate confounder traits are not necessarily true confounders, but are very likely to be proxies for a confounding parental environment/trait. This is also subject to further future research.

6. In Section 3.3.2 you say "Thus, we obtained a massively attenuated causal effect of BMI on EDU, when childhood BMI is used as an exposure." I am wondering if it makes sense to directly compare the magnitude of the effect estimates for adult and childhood BMI, given that childhood BMI was proxied by a three level categorical variable of relative body size. Is a one standard deviation increase in this categorical variable really comparable to a one standard deviation increase in BMI? (If BMI were approximately normally distributed, two individuals could differ by 1 SD of BMI but still both be in the middle third of the distribution and therefore have the same level of "relative body size").

Thank you, this is an excellent point, and we ran simulations to address this problem. We simulated a PRS to explain 10% of childhood BMI and added Gaussian noise to generate childhood BMI values. Individuals were then split into three categories, matching the proportion of plumper and skinnier subjects in the UK Biobank data. We then normalised this trichotomized phenotype to have a variance of 1 (mimicking our original analysis). Both the real and the trichotomized childhood BMI was regressed onto the PRS. Next, we simulated a continuous EDU score with true childhood BMI having a small (-0.1) causal effect on it. Finally, we ran MR for both the trichotomized and true childhood BMI on EDU and compared the magnitudes of the causal effects of 100 different runs (figure below).

Figure 3 - Causal effect estimate of childhood BMI and trichotomized BMI on education. True causal effect of BMI on EDU is -0.1.

As seen in the results above, the causal effect estimates of BMI and trichotomized BMI (tBMI) on EDU are comparable, with a slight (10%) increase of the average causal effect of tBMI in comparison to BMI's effect. This means that using a trichotomised version of childhood BMI may have led to a slight overestimation of the causal effect, therefore the true childhood BMI on EDU effect may be even smaller than the estimated one. Furthermore, we see that 1 SD change in tBMI is equivalent to 0.9 SD change in BMI, assuring us of the robustness of our results and data used.

7. It would be good to add labels to Table 2 for K-means and MR-Clust. It would also be nice to have a little of the interpretation of this analysis in the results section rather than in the discussion, since it is not immediately clear from methods or results what the point of this analysis is.

We added an extra cell to indicate that MR-Clust clusters are at the rows and that the columns are PWC-MR clusters. We added an extra sentence to highlight that low agreement between the two types of clustering.

Reviewer #1 (Remarks to the Author):

The authors have thoroughly and adequately responded to my comments.

Reviewer #2 (Remarks to the Author):

I'd like to commend the authors on a nice revision.

My primary remaining issue is that it is not very clear exactly what conclusions we can draw from the clustering analysis about the effect of BMI on EDU.

I think the proposition in the discussion that the large estimate of the BMI → EDU effect from population-based GWAS is related to dynastic effects is reasonable. It could also be a result of uncorrected population stratification, which I think would yield the same pattern. In either case, this conclusion is supported by the MVMR analysis which selected three variables that could reasonably be expected to be strongly related to socio-economic position and/or ancestry. It may be that associations with these traits primarily capture associations with SEP or associations with ancestry and so adjusting for them removes that bias. I think it is worth pointing out that the estimate you get from the MVMR analysis is very close to the estimate from the sib-pair analysis.

What I think is lacking is the connection between the cluster analysis and these two other analyses. If correct, it would be nice to state explicitly that you think the cluster 2 estimate is the best estimate of the causal effect with a justification. If that isn't correct, that would also be good to clarify as it seems implied in some of the wording that this is the case.

The other missing piece is to make explicit how the cluster analysis is an improvement over the MVMR analysis, if you think that it is.

One minor comment - it would be nice in the labeling of Figure 4 to make it clear that the clusters for the different traits contain different variants and are not comparable. You could use different letters for example.

Reviewer #3 (Remarks to the Author):

I have been asked to participate in the review process of this study. The authors of this study proposed a new method, PWC-MR, which leverages information from multiple traits for clustering of IVs to reveal different mechanisms between an exposure and an outcome. The method is applied to study the case of BMI and EDU. I appreciate the authors' innovative approach of incorporating external information to uncover underlying mechanisms linking the exposure and the outcome. I do have some concerns related to your responses to Reviewer #1's comments.

(Comment 1 of review #1) In Comment 1, Reviewer #1 expresses doubt regarding the efficiency of the proposed method in addressing confounding bias compared to other methods. The authors attempt to address this concern by suggesting that the estimates from the PWC-MR method and other methods may be comparable based on a difference in p-values. However, the calculation of this p-value difference remains unclear to me. Moreover, the authors highlight a unique feature of the PWC-MR method, which does not rely on within-family GWAS. Although PWC-MR can yield multiple mechanism-specific estimates using population-based GWAS, it does not provide a means to determine which estimate is closest to the true causal effect or which group of IVs is less susceptible to MR violations.

(Comment 2 of reviewer #1) I think the reviewer and I share concerns regarding the potential

sensitivity of the clustering results to the choice of model selection criteria and clustering methods. It is difficult to evaluate which choice yields more meaningful results.

(Comment 7 of reviewer #1) I did not observe U_G or U in Figure 1 as mentioned in your response.

(Comment 13 of reviewer #1) I think the current response to this question of reviewer #1 is not satisfactory. The authors should directly address whether the proposed method is comparable to a theoretically driven selection of curated instruments for mechanism-specific causal effect estimates. Including an illustration using metabolism-related IVs for BMI, as suggested by Reviewer #1, would be a more convincing demonstration.

REVIEWER COMMENTS & RESPONSES NCOMMS-23-11823A - PART II

Overall responses: We sincerely thank the reviewers for going through our updated manuscript and taking the time to give us feedback. We are glad that the changes made in the main manuscript and the supplementary material were well received, and we hope that the additional changes done in response to their latest comments are satisfactory.

Reviewer #1 (Remarks to the Author):

The authors have thoroughly and adequately responded to my comments.

We thank the reviewer for their time and helpful feedback!

Reviewer #2 (Remarks to the Author):

I'd like to commend the authors on a nice revision.

We would also like to thank the reviewer for their helpful comments that aided in improving the clarity of the manuscript and the strength of our results. We did our best to adequately address all remaining comments.

My primary remaining issue is that it is not very clear exactly what conclusions we can draw from the clustering analysis about the effect of BMI on EDU.

We appreciate your comment regarding the conclusions of our work, and we would like to clarify that this information is already presented in the manuscript, specifically in the discussion quoted below:

*"For our exposure, BMI, **six distinct clusters** were obtained through optimal K-means clustering. These clusters had well-defined trait enrichments, with clusters matching SEP-related, substrate, and body measurement traits. Estimating individual causal effects of each cluster on EDU as an outcome revealed **heterogeneous causal effect estimates** which allowed us to further strengthen our suspicion that the **MR estimate for the causal effect of BMI on EDU is upward biased** when using population-based SNP effect size estimates **due to confounding**.*

...

*Our results have **two major implications**: 1) The lean-mass-related IV cluster indicated a more plausible, close to zero causal effect of BMI on EDU. 2) We revealed that the SEP-related IVs leading to an apparent, sizeable negative effect of BMI on EDU, possibly overestimating the true underlying causal effect.*

...

*To conclude, we would like to reiterate that although we believe that the classical MR estimate based on population GWAS leads to an **overestimation of the BMI-EDU causal effect**, we are uncertain whether **this effect is exactly zero, or is just strongly attenuated**.*

*The **attenuated estimate provided by our PWC-MR approach (cluster #2) is compatible with both the estimate based on sib-regression summary statistics (P-values difference = 0.161) and the MVMR estimate (p -diff = 0.476), all of which are***

based on adulthood phenotypes. However, the estimate obtained for childhood BMI is slightly more attenuated than that of the PWC-MR method (p-diff 0.024)."

We have now slightly extended the concluding statements of the Discussion to:

"To conclude, we found that the classical MR estimate based on population GWAS leads to an overestimation of the BMI-EDU causal effect and identified an lean-mass-specific subgroup of IVs that, we believe, yield a much more reliable causal effect estimate. Still, we are uncertain whether this effect is exactly zero, or is just strongly attenuated. Our analysis also revealed that the unrealistically large standard MR estimate was driven by IVs that likely violate the pleiotropy assumption via being also linked to SEP."

I think the proposition in the discussion that the large estimate of the BMI -> EDU effect from population-based GWAS is related to dynastic effects is reasonable. It could also be a result of uncorrected population stratification, which I think would yield the same pattern. In either case, this conclusion is supported by the MVMR analysis which selected three variables that could reasonably be expected to be strongly related to socio-economic position and/or ancestry. It may be that associations with these traits primarily capture associations with SEP or associations with ancestry and so adjusting for them removes that bias. I think it is worth pointing out that the estimate you get from the MVMR analysis is very close to the estimate from the sib-pair analysis.

Thank you for pointing this out, we extended our discussion of the MVMR results to include this:

"The attenuated estimate provided by our PWC-MR approach (cluster #2) is compatible with both the estimate based on sib-regression summary statistics (P-values difference = 0.161) and the MVMR estimate (p-diff = 0.476), all of which are based on adulthood phenotypes. However, the estimate obtained for childhood BMI is slightly more attenuated than that of the PWC-MR method (p-diff 0.024)."

What I think is lacking is the connection between the cluster analysis and these two other analyses. If correct, it would be nice to state explicitly that you think the cluster 2 estimate is the best estimate of the causal effect with a justification. If that isn't correct, that would also be good to clarify as it seems implied in some of the wording that this is the case.

Addressing the first part of the comment about the connection between our findings from the cluster analysis and both the the sib-regression and MVMR results: we note that the causal effect estimate of cluster #2 is similar to both (not significantly different from) the sib-regression causal effect estimate and the attenuated BMI causal effect estimate in MVMR conditional on the candidate confounder traits, as mentioned in the last section of the discussion:

"The attenuated estimate provided by our PWC-MR approach (cluster #2) is compatible with both the estimate based on sib-regression summary statistics (P-values difference = 0.161) and the MVMR estimate (p-diff = 0.476), all of which are based on adulthood phenotypes."

Furthermore, in the Results section (page 9, lines 225-228), we mentioned the link between the MVMR candidate confounders, and their enrichment ratio to the various SNPs in the six clusters:

"Matching the 19 confounder traits from this analysis to their respective ERs across the six clusters from the previous analysis revealed higher ERs in cluster #1 and cluster #4 (see Supplementary Figure S6), which was associated with SEP-related traits."

However, we have updated this sentence to include that both clusters #1 and #4 had the most negative causal effects on EDU, fitting the narrative that these candidate confounders were leading to an overestimation of the causal effect.

As for the second part of the comment, we made it more explicit that cluster #2 gives the most likely valid estimate:

“To conclude, we found that the classical MR estimate based on population GWAS leads to an overestimation of the BMI-EDU causal effect and identified an obesity-specific subgroup of IVs (cluster #2) that, we believe, yields a much more reliable causal effect estimate.”

The other missing piece is to make explicit how the cluster analysis is an improvement over the MVMR analysis, if you think that it is.

We thank the reviewer for this excellent point which was missing from our discussion. We have since updated our discussion to include that our PWC methodology offers an advantage over MVMR where it can ascertain different sets of SNPs that can be associated to different mechanisms and potentially reveal heterogeneous causal effects on certain outcome traits. Whereas in MVMR, you get a single homogeneous estimate of the causal effect a trait has on an outcome, that is unbiased given that other confounder/pleiotropic traits are being accounted for. Furthermore, our informative SNP clustering also offers us a biological insights into the potential underlying mechanisms and pathways that a trait may have, which we can then use to further investigate heterogeneous causal effects, perhaps through looking up loci regions for certain clusters and checking their eQTL enrichment across various tissues.

“... PWC-MR offers a number of features that compliment other approaches: ... (iii) in contrast to MVMR, which estimates a single causal effect, PWC-MR provides multiple causal effect estimates, some of which may reflect confounder effects, and others heterogeneous mechanisms of action, overall revealing biological insight that can be used in follow-up research.”

One minor comment - it would be nice in the labeling of Figure 4 to make it clear that the clusters for the different traits contain different variants and are not comparable. You could use different letters for example.

We agree with the reviewer that our labeling may be a source of confusion, and have thus updated the figure labels (specifically the x-labels of panels a, b, and c) to indicate which trait the SNP clusters belong to:

Forest plot of IVW causal effect estimate on outcome using either all exposure IVs (All) or cluster-specific IVs (C1..C4..C6). Panel a shows causal effect estimates of adult BMI on EDU, panel b proxy of childhood BMI (cBMI) on EDU, and panel c adult BMI on SBP. Horizontal error bars represent the 95% confidence interval. The blue vertical line represents the causal effect estimated using all BMI/cBMI IVs. Box sizes of clusters represent the proportion of the number of IVs in each cluster to the total.

Reviewer #3 (Remarks to the Author):

I have been asked to participate in the review process of this study. The authors of this study proposed a new method, PWC-MR, which leverages information from multiple traits for clustering of IVs to reveal different mechanisms between an exposure and an outcome. The method is applied to study the case of BMI and EDU. I appreciate the authors' innovative approach of incorporating external information to uncover underlying mechanisms linking the exposure and the outcome. I do have some concerns related to your responses to Reviewer #1's comments.

We thank the reviewer for their time and comments and we made every effort to address their concerns adequately.

(Comment 1 of review #1) In Comment 1, Reviewer #1 expresses doubt regarding the efficiency of the proposed method in addressing confounding bias compared to other methods. The authors attempt to address this concern by suggesting that the estimates from the PWC-MR method and other methods may be comparable based on a difference in p-values. However, the calculation of this p-value difference remains unclear to me. Moreover, the authors highlight a unique feature of the PWC-MR method, which does not rely on within-family GWAS. Although PWC-MR can yield multiple mechanism-specific estimates using population-based GWAS, it does not provide a means to determine which estimate is closest to the true causal effect or which group of IVs is less susceptible to MR violations.

We apologise for any ambiguity in our previous response, the p-value previously calculated is that of a t-test to investigate the difference in means:

$$t = \frac{b_1 - b_2}{\sqrt{SE_1^2 + SE_2^2}}, \text{ where we calculate the p-value by using the normal approximation:}$$

$$P = 2 * \phi(-|t|)$$

Here, b_1 represent the causal effect estimate of method 1 (e.g.: PWC-MR) and b_2 that of the other method (e.g.: sib-regression MR). SE represents the standard error of the estimate for each method.

We also thank the reviewer for mentioning the second point which is very pertinent. Indeed, from only the various cluster-specific causal effect estimates, one cannot determine which represents the true, or most likely to be true causal effect estimate. In our case, for BMI as an exposure and its estimated causal effect on EDU, we were fortunate to have two other sources from which the estimated causal effects were more reliable: the sib-regression MR estimate, and the attenuated MVMR causal effect estimate. This helped us pinpoint the likely correct cluster-specific causal effect estimate which was less susceptible to violations/confounding. Additionally, the enrichment ratio of various traits across the 6 clusters provided an additional level of confidence given that cluster #2 was enriched for body-measurement traits, specifically lean-mass measurement traits. But indeed, such additional piece of evidence may not always be available, which we added as a new limitation:

“Fifth, while for the BMI-EDU relationship we had several lines of evidence pinpointing cluster #2 as the one yielding the most likely correct causal effect estimate, in general, we might not be able to decide which cluster(s) provide biologically meaningful causal effect estimate(s) and which ones may be linked to confounders.”

(Comment 2 of reviewer #1) I think the reviewer and I share concerns regarding the potential sensitivity of the clustering results to the choice of model selection criteria and clustering methods. It is difficult to evaluate which choice yields more meaningful results.

We appreciate your comment regarding the clustering and model choice criteria that we used. We have used **different model selection criteria and additional number of clusters to show that the essence of the results does not change. We very robustly always get a body shape/weight-related cluster and other clusters (some containing SEP-related IVs)**. We paste here the detailed explanation of the different sensitivity analyses we performed to prove this point, currently added to Supplementary 1.1:

“In order to test for multiple model selection criteria, we tested for the optimal cluster number using both AIC (as shown in the manuscript) and Bayesian information criterion (BIC).

Using BIC, we end up with 2 clusters being the optimal for BMI SNPs with heterogeneous causal effects on EDU (cluster 1 = -0.13 (6.61E-16), cluster 2 = -0.34 (6.15E-44)), and their enrichment reflects a clear distinction between enrichment for lean-mass and body related traits in cluster 1 and a mixed bag of trait enrichment for cluster 2 including lung/height/blood and SES-proxy traits.

This result is due to BIC introducing a stronger penalty term, $k \times \log(n)$, where k is the number of clusters and n corresponds to the number of (independent) samples used.

However, in our case n represents the number of traits, which are highly correlated. Also, the more traits are used to cluster the SNPs, the more clusters we expect to obtain as they

allow for a more fine-grain resolution of the underlying biological mechanisms. For these reasons, we do not believe that BIC is an appropriate measure to quantify clustering fit in this situation.

Therefore, the BIC-based selection of optimal cluster number does not alter the main message/result, and only leads to coarser grain clusters.

Figure 1 - Left: Causal effect estimates of BIC-clustered BMI SNPs on educational attainment.

On the other hand, we also tried to forcibly increase the number of clusters to 8 in the hopes of achieving more distinction in enrichment. We observed similar heterogeneous causal effects on EDU, where the smallest and largest effects were from clusters enriched for lean mass and SES-related traits respectively. As for the rest of the clusters, another 2 were strongly enriched for food supplements and a mix of height/blood/lung measurement traits, another was enriched for a mix of diseases and three other clusters had low enrichments for miscellaneous traits.

Right: Causal effect estimates of clustered (forced 8 clusters) BMI SNPs on educational attainment.”

(Comment 7 of reviewer #1) I did not observe U_G or U in Figure 1 as mentioned in your response.

We apologise for any confusion this may have caused, in the revision of the manuscript we moved the previous 'Figure 1 - Panel a' to 'Supplementary Figure 9'.

(Comment 13 of reviewer #1) I think the current response to this question of reviewer #1 is not satisfactory. The authors should directly address whether the proposed method is comparable to a theoretically driven selection of curated instruments for mechanism-specific causal effect estimates. Including an illustration using metabolism-related IVs for BMI, as suggested by Reviewer #1, would be a more convincing demonstration.

We thank the reviewer for their comment, and we have attempted our best to address this comment and the original one in an applied manner. To this end, we:

- 1) selected all the genes that are involved in the metabolism pathway (KEGG:hsa01100 - Metabolic pathways - Homo sapiens (human)), which turned out to be 1543.
- 2) selected genes in a 1Mb proximity to our 324 BMI IVs, of which there were 358.
- 3) then searched for the overlap between these two sets of genes, to trace back the metabolism-related IVs for BMI. There were 12 genes in the intersection.
- 4) using the 12 BMI IVs associated with these 12 overlapped genes, we ran MR to estimate the causal effect on EDU, and obtained an IVW estimate of **-0.24 (p-value = 5.32E-4)**, see table 1 below
- 5) also checked the spread of the SNPS across the 6 clusters and found them more or less evenly spread

Table 1: Causal effect estimate of BMI SNPs curated for metabolism pathway on EDU

Method	nSNP	b	SE	p-val
MR Egger	12	0.0603	0.3110	0.8501
Weighted median	12	-0.2311	0.0704	1.03E-03
Inverse variance weighted	12	-0.2433	0.0702	5.32E-04
Simple mode	12	-0.2065	0.1043	7.32E-02
Weighted mode	12	-0.1912	0.0852	0.0464

Table 2: Spread of BMI SNPs curated for metabolism pathways across the 6 clusters

C1	C2	C5	C6
3	5	3	1

These results show that curating SNPs by the metabolism pathway does not lead to a causal effect significantly different from any 12 randomly chosen SNP's causal effect. Furthermore the SNPs selected this way do not seem to overlap significantly any of our 6 clusters.

Other pathways or mechanisms that can be used to curate BMI SNPs include appetite regulation, lipolysis or adipogenesis, all extremely specific aspects of BMI.

Thus, we performed the same analysis searching in the KEGG database for genes involved in the lipolysis (hsa04923) and adipogenesis pathways (hsa04920) to obtain the following results:

Table 3: Causal effect estimate of BMI SNPs curated for lipolysis pathway on EDU

Method	nsnp	b	SE	p-val
Inverse variance weighted	2	-0.2092	0.2671	0.4335

Table 4: Spread of BMI SNPs curated for lipolysis across the 6 clusters

C2	C4
1	1

Table 5: Causal effect estimate of a BMI SNP curated for adipogenesis pathway on EDU, found in *cluster 5*

Method	nsnp	b	SE	p-val
Wald ratio	1	-0.2396	0.2214	0.2791

These results also show that curating SNPs by the either lipolysis or adipogenesis pathways does not reveal an attenuated causal effect between BMI and EDU compared to using all BMI SNPs.

Reviewer #2 (Remarks to the Author):

All of my comments have been addressed satisfactorily. Congratulations to the authors on a nice article.

Reviewer #3 (Remarks to the Author):

Thank you for all the efforts to address my comments